# Evolutionary transcriptomics implicates *HAND2* in the origins of implantation and regulation of gestation length

**Mirna Marinić[1†], Katelyn Mika[1†], Sravanthi Chigurupati[1‡], Vincent J Lynch[2]\***

[1]Department of Human Genetics, University of Chicago, Chicago, United States; [2]Department of Biological Sciences, University at Buffalo, Buffalo, United States

**Abstract** The developmental origins and evolutionary histories of cell types, tissues, and organs contribute to the ways in which their dysfunction produces disease. In mammals, the nature, development and evolution of maternal-fetal interactions likely influence diseases of pregnancy. Here we show genes that evolved expression at the maternal-fetal interface in Eutherian mammals play essential roles in the evolution of pregnancy and are associated with immunological disorders and preterm birth. Among these genes is *HAND2*, a transcription factor that suppresses estrogen signaling, a Eutherian innovation allowing blastocyst implantation. We found dynamic *HAND2* expression in the decidua throughout the menstrual cycle and pregnancy, gradually decreasing to a low at term. HAND2 regulates a distinct set of genes in endometrial stromal fibroblasts including *IL15*, a cytokine also exhibiting dynamic expression throughout the menstrual cycle and gestation, promoting migration of natural killer cells and extravillous cytotrophoblasts. We demonstrate that *HAND2* promoter loops to an enhancer containing SNPs implicated in birth weight and gestation length regulation. Collectively, these data connect *HAND2* expression at the maternal-fetal interface with evolution of implantation and gestational regulation, and preterm birth.

**\*For correspondence:**
vjlynch@buffalo.edu

**Present address:** [†]Department of Organismal Biology and Anatomy, University of Chicago, Chicago, United States; [‡] AbbVie, North Chicago, United States

**Competing interests:** The authors declare that no competing interests exist.

## Introduction

The ontogeny and evolutionary history of cell types, tissues, and organ systems, as well as the life histories of organisms bias the ways in which dysfunctions in those systems underlie disease (*Varki, 2012*). Thus, a mechanistic understanding of how cells, tissues, and organs evolved their functions, and how organism's life histories influence them, may provide clues to the molecular etiologies of disease. The most common way of utilizing evolutionary information to characterize the genetic architecture of disease is to link genetic variation within a species to phenotypes using quantitative trait loci (QTL) or genome-wide association studies (GWAS). An alternative approach is to identify fixed genetic differences between species that are phylogenetically correlated with different disease relevant phenotypes. While the risk of cancer increases with the age of an individual, for example, the prevalence of cancer types varies by species (*Abegglen et al., 2015*), likely because of differences in genetic susceptibility to specific cancers, structure of organ and tissue systems, and life exposures to carcinogens (*Varki and Varki, 2015*). Similarly, the risk of cardiovascular disease (CVD) increases with age across species, but the pathophysiology of CVD can differ even between closely related taxa such as humans, in which CVD predominantly results from coronary artery atherosclerosis, and the other Great Apes (Hominids), in which CVD is most often associated with interstitial myocardial fibrosis (*Varki et al., 2009*).

Extant mammals span major stages in the origins and diversification of pregnancy, thus a mechanistic understanding of how pregnancy originated and diverged may provide unique insights into the ontogenetic origins of pregnancy disorders. The platypus and echidna (Monotremes) are oviparous, but the embryo is retained in the uterus for 10–22 days, during which the developing fetus is

nourished by maternal secretions delivered through a simple placenta, prior to the laying of a thin, poorly mineralized egg that hatches in ~2 weeks (*Hill, 1936*). Live birth (viviparity) evolved in the stem-lineage of Therian mammals, but Marsupials and Eutherian ('Placental') mammals have dramatically different reproductive strategies. In Marsupials, pregnancies are generally short (~25 days) and completed within the span of a single estrous cycle (*Renfree and Shaw, 2001*; *Renfree, 2010*). Eutherians, in contrast, evolved a suite of traits that support prolonged pregnancies (up to 670 days in African elephant), including an interrupted estrous cycle, which allows for gestation lengths longer than a single reproductive cycle, maternal-fetal communication, maternal recognition of pregnancy, implantation of the blastocyst and placenta into uterine tissue, differentiation (decidualization) of endometrial stromal fibroblasts (ESFs) in the uterine lining into decidual stromal cells (DSCs), and maternal immunotolerance of the antigenically distinct fetus, that is the fetal allograft (*Guleria and Pollard, 2000*; *Moffett and Loke, 2004*; *Erlebacher, 2013*).

Gene expression changes at the maternal-fetal interface underlie evolutionary differences in pregnancy (*Hou et al., 2012*; *Lynch et al., 2015*; *Armstrong et al., 2017*), and thus likely also pathologies of pregnancy such as infertility, recurrent spontaneous abortion (*Kosova et al., 2015*), preeclampsia (*Elliot, 2017*; *Arthur, 2018*; *Varas Enriquez et al., 2018*), and preterm birth (*Plunkett et al., 2011*; *Swaggart et al., 2015*; *LaBella, 2019*). Here, we assembled a collection of gene expression data from the pregnant/gravid maternal-fetal interface of tetrapods and used evolutionary methods to reconstruct gene expression changes during the origins of mammalian pregnancy. We found that genes that evolved to be expressed at the maternal-fetal interface in the Eutherian stem-lineage were enriched for immune functions and diseases, as well as preterm birth. Among the recruited genes was the transcription factor *Heart- and neural crest derivatives-expressed protein 2* (*HAND2*), which plays essential roles in neural crest development (*Srivastava et al., 1997*), cardiac morphogenesis (*Srivastava et al., 1997*; *Shen et al., 2010*; *Tamura et al., 2013*; *Lu et al., 2016*; *Sun et al., 2016*), and suppressing estrogen signaling during the period of uterine receptivity to implantation (*Huyen and Bany, 2011*; *Li et al., 2011*; *Shindoh et al., 2014*; *Fukuda et al., 2015*; *Mestre-Citrinovitz et al., 2015*; *Murata et al., 2019*; *Šućurović et al., 2020*). We determined that *HAND2* expression at the first trimester maternal-fetal interface was almost entirely restricted to cell types in ESF lineage and is regulated by multiple transcription factors that control progesterone responsiveness. Moreover, the *HAND2* promoter loops to an enhancer with single-nucleotide polymorphisms (SNPs) that have been implicated by GWAS in the regulation of gestation length (*Warrington et al., 2019*; *Sakabe et al., 2020*). Furthermore, we showed that HAND2 regulates interleukin 15 (*IL15*) expression in ESFs, and that ESF-derived IL15 influences the migration of natural killer and trophoblast cells. These data suggest that HAND2 and IL15 signaling played an important role in the evolution of implantation and regulation of gestation length.

## Results

### Genes that evolved endometrial expression in Eutherian mammals are enriched in immune functions

We previously used comparative transcriptomics to reconstruct the evolution of gene expression at the maternal-fetal interface during the origins of mammalian pregnancy (*Lynch et al., 2015*). Here, we assembled a collection of new and existing transcriptomes from the pregnant/gravid endometria of 15 Eutherian mammals, 3 Marsupials, 1 Monotreme (platypus), 2 birds, 5 lizards, and 1 amphibian (*Figure 1A* and *Figure 1—source data 1*). The complete dataset includes expression information for 21,750 genes from 27 species. Next, we transformed continuous transcript abundance estimates values into discrete character states such that genes with Transcripts Per Million (TPM) $\geq$ 2.0 were coded as expressed (state = 1), genes with TPM < 2.0 were coded as not expressed (state = 0), and genes without data in specific species were coded as missing (state = ?). We then used parsimony to reconstruct ancestral transcriptomes and trace the evolution of gene expression gains (0 → 1) and losses (1 → 0) in the endometrium (*Figure 1—source data 2*).

We identified 958 genes that most parsimoniously evolved endometrial expression in the Eutherian stem-lineage, including 149 that unambiguously evolved endometrial expression (*Figure 1A* and *Figure 1—source data 3*). These 149 genes were significantly enriched in pathways related to

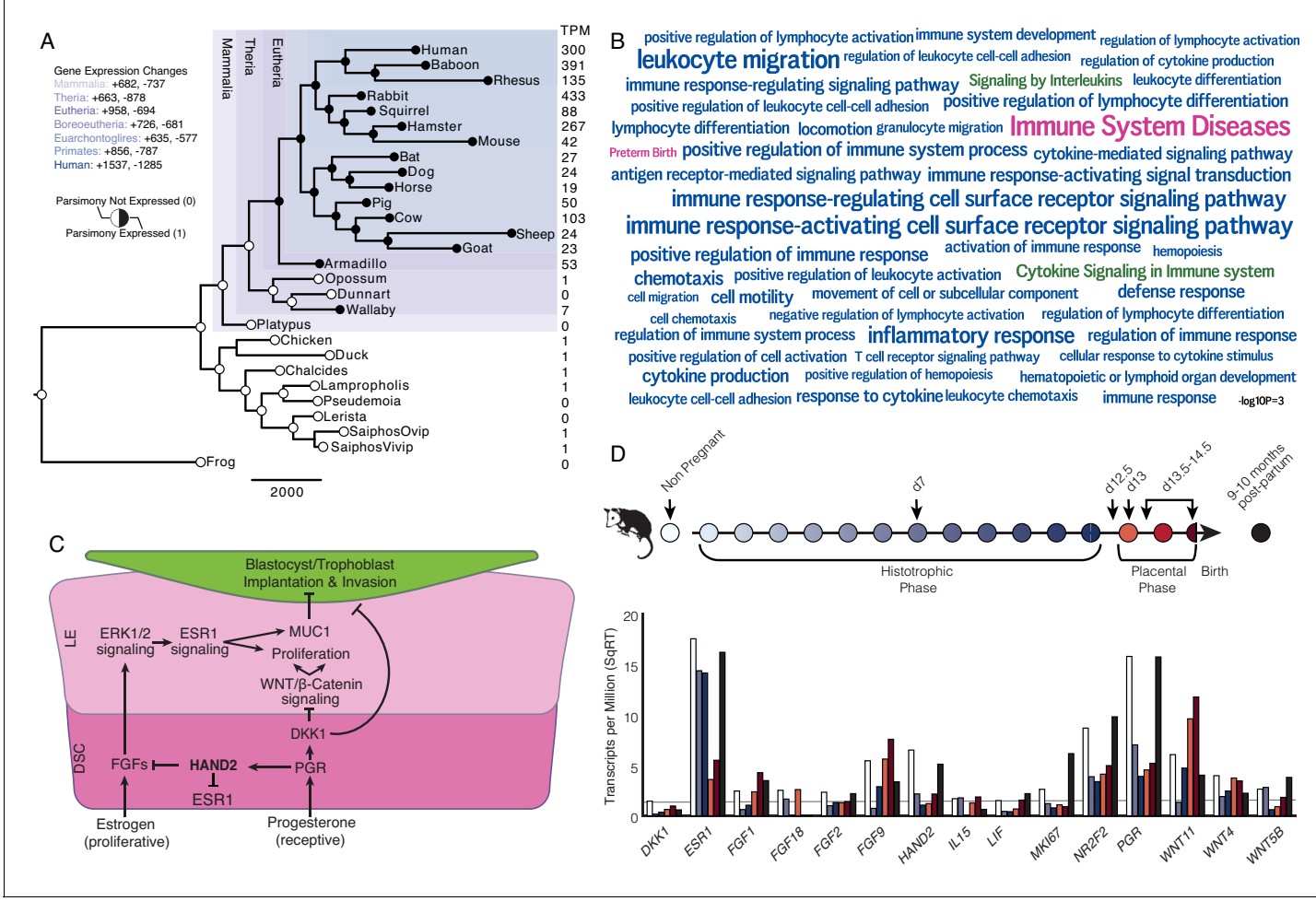

**Figure 1.** Recruitment of HAND2-mediated anti-estrogenic signaling in the Eutherian endometrium. (A) Evolution of *HAND2* expression at the maternal-fetal interface. Amniotes phylogeny with horizontal branch lengths drawn proportional to the number of gene expression changes inferred by parsimony (most parsimonious reconstruction). Circles indicate *HAND2* expression in extant species and ancestral reconstructions. Black, expressed (state = 1). White, not expressed (state = 0). Inset legend shows the number of most gene expression changes from the root node to human (+ = gene expression gained; - = gene expression lost). Numbers to the right indicate *HAND2* expression in TPM for each respective species. (B) WordCloud of biological pathways (green), human disease phenotypes (pink), and biological process gene ontology terms (blue) enriched among 149 unambiguously recruited genes in the Eutherian stem-lineage. Term size is shown scaled to -log10 p-value (see inset scale). (C) Cartoon model of estrogen signaling and HAND2-mediated anti-estrogenic signaling in the endometrium. The estrogen-mediated signaling network is suppressed by progesterone through the activation of HAND2 and antagonists of canonical WNT/β-catenin-mediated signaling pathways such as DKK1. In the proliferative phase of the reproductive cycle, estrogen acts through ESR1 in stromal cells to increase the production of fibroblast growth factors (FGFs), which serve as paracrine signals leading to sustained proliferation of epithelial cells. Active estrogen signaling maintains epithelial expression of Mucin 1 (MUC1), a cell surface glycoprotein that acts as a barrier to implantation. During the receptive phase of the cycle, however, progesterone induces HAND2 and DKK1 expression in the endometrial stroma, inhibiting production of FGFs, suppressing epithelial proliferation and antagonizing estrogen-mediated expression of MUC1, thereby promoting uterine receptivity to implantation. DSC = decidual stromal cells, LE = luminal epithelium. (D) Gene expression time course through opossum pregnancy. Upper, schematic of gestation length in *Monodelphis domestica* in which the histotrophic phase lasts from day 1 to day 12, hatching occurs on day 12.5, the placental phase lasts from day 13 to day 14.5, and birth occurs on day 14.5. Lower, data shown as square root (SqRT) transformed TPM. The TPM = 2 expression cutoff is shown as a horizontal gray line. *M. domestica* RNA-Seq data from *Lynch et al., 2015*; *Hansen et al., 2016*; *Griffith et al., 2017*; *Griffith et al., 2019*.

The online version of this article includes the following source data and figure supplement(s) for figure 1:

**Source data 1.** Species and gene expression information.
**Source data 2.** Binary encoded endometrial gene expression dataset.
**Source data 3.** Genes (HUGO gene names) that unambiguously evolved endometrial expression in the Eutherian stem-lineage ('Recruited Genes').
**Source data 4.** Top 100 pathways (Wikipathway, Reactome, KEGG) in which Eutherian recruited genes are enriched.
**Source data 5.** Top 100 human phenotype (disease) ontology terms in which Eutherian recruited genes are enriched.
**Source data 6.** Top 100 biological process gene ontology (GO) terms in which Eutherian recruited genes are enriched.

*Figure 1 continued on next page*

*Figure 1 continued*

**Source data 7.** RNA-Seq data from opossum endometrial samples.

**Source data 8.** Database of genes implicated in preterm birth.

**Figure supplement 1.** The unpaired mean difference in *HAND2* expression between non-Gravid and Gravid wallaby samples is −8.6 [95.0% CI: −10.7 – −6.46].

**Figure supplement 1—source data 1.** Uterine gene expression data (in TPM) from four Gravid (orange - Wal38, Wal40, Wal57, and Wal59) and four non-Gravid (blue - Wal39, Wal41, Wal58, and Wal60) tammar wallabies.

**Figure supplement 2.** Immunohistochemistry showing phosphorylated ESR1 (pESR1), phosphorylated MAPK1/2 (pERK1/2) and MUC1 expression in paraffin-embedded 12.5d pregnant *Monodelphis domestica* endometrium compared to control (IgG).

the immune system (*Figure 1B* and *Figure 1—source data 4–6*), although only two pathways were enriched at False Discovery Rate (FDR) $\leq$ 0.10, namely, 'Cytokine Signaling in Immune System' (hypergeometric p = 1.97×10$^{-5}$, FDR = 0.054) and 'Signaling by Interleukins' (hypergeometric p = 4.09×10$^{-5}$, FDR = 0.067). Unambiguously recruited genes were also enriched in numerous human phenotype ontology terms but only two, 'Immune System Diseases' (hypergeometric p = 3.15×10$^{-8}$, FDR = 2.52×10$^{-4}$) and 'Preterm Birth' (hypergeometric p = 4.04×10$^{-4}$, FDR = 8.07×10$^{-4}$), were enriched at FDR $\leq$ 0.10. In contrast, these genes were enriched in numerous biological process gene ontology (GO) terms at FDR $\leq$ 0.10, nearly all of which were related to regulation of immune system, including 'Leukocyte Migration' (hypergeometric p = 1.17×10$^{-7}$, FDR = 1.29×10$^{-3}$), 'Inflammatory Response' (hypergeometric p = 8.07×10$^{-7}$, FDR = 2.21×10$^{-3}$), and 'Cytokine-mediated Signaling Pathway' (hypergeometric p = 2.18×10$^{-5}$, FDR = 0.013).

## Recruitment of *HAND2* and anti-estrogenic signaling in Eutherians

Among the genes that unambiguously evolved endometrial expression in the Eutherian stem-lineage was the basic helix-loop-helix family transcription factor *Heart- and neural crest derivatives-expressed protein 2* (*HAND2*). *HAND2* plays an essential role in mediating the anti-estrogenic action of progesterone and the establishment of uterine receptivity to implantation (*Figure 1C*; *Huyen and Bany, 2011*; *Li et al., 2011*; *Fukuda et al., 2015*; *Mestre-Citrinovitz et al., 2015*), suggesting that the silencing of estrogen signaling during the window of implantation is a derived trait in Eutherian mammals. Notably, this silencing is not observed in the pregnant endometrium of Marsupials such as the brush-tailed possum (*Trichosurus vulpecula*) (*Young and McDonald, 1982*; *Curlewis and Stone, 1987*) and tammar wallaby (*Macropus eugenii*) (*Renfree and Blanden, 2000*).

To investigate further, we used the short-tailed opossum (*Monodelphis domestica*) as a model of pregnancy in Marsupials. *M. domestica* lacks implantation and thus is a good representative of pregnancy in the Therian common ancestor, in contrast to other Marsupials, such as the tammar wallaby, which have derived traits such as delayed ovulation, independently evolved maternal recognition of pregnancy and expression of *HAND2* at low levels during gravidity (*Figure 1A* and *Figure 1—figure supplement 1—source data 1*, *Figure 1—figure supplement 1*). Using existing RNA-Seq data from short-tailed opossum endometria, we analyzed a time course consisting of day 7 (during the histotrophic phase), day 12.5 (just after hatching and during the transition from the histotrophic to the placental phase), day 13 (early placental phase), day 13.5–14.5 (during the late placental to early parturition phase), and 9–10 month post-partum, as well as non-pregnant control (*Figure 1D*; *Lynch et al., 2015*; *Hansen et al., 2016*; *Griffith et al., 2017*; *Griffith et al., 2019*). We found that *HAND2* was abundantly expressed in the non-pregnant endometrium and down-regulated throughout the histotrophic phase, reaching a low (TPM < 2) at 12.5d, and subsequently increasing in expression in the 13.5-14d samples near term. *ESR1*, *FGF2*, *FGF9*, *FGF18* and several *WNT* genes that stimulate proliferation of the luminal epithelia were abundantly expressed (*Figure 1D* and *Figure 1—source data 7*), consistent with persistent estrogen signaling during pregnancy (*Renfree and Blanden, 2000*). Both *HAND2* and *ESR1* decrease during gestation (*Figure 1D*), suggesting that the inhibition of estrogen signaling by *HAND2* seen in Eutherians does not occur in opossum endometrium – if it did, one would expect *ESR1* expression to increase as *HAND2* expression decreased. Similarly, *HAND2* is down-regulated during pregnancy in the tammar wallaby (*Figure 1—figure supplement 1—source data 1*, *Figure 1—figure supplement 1*). Immunohistochemistry (IHC) on opossum endometrial sections from day 12.5 pregnant endometrium stained strongly for estrogen receptor alpha (*ESR1*; ERα) phosphorylated at serine 118, a mark of transcriptionally active ERα

(*Kato et al., 1995*), as well as phosphorylated ERK1/2, and MUC1 (*Figure 1—figure supplement 2*), which is also consistent with active estrogen signaling.

## *HAND2* is expressed in endometrial stromal fibroblast lineage cells

To determine which cell types at the human maternal-fetal interface express *HAND2*, we used previously generated single-cell RNA-Seq (scRNA-Seq) data from the first trimester decidua (*Vento-Tormo et al., 2018*). *HAND2* expression was almost entirely restricted to cell populations in the endometrial stromal fibroblast lineage (see Materials and methods for cell type naming convention), with particularly high expression in ESF2s and DSCs (*Figure 2A*). Interestingly, while it is generally thought that ESFs are not present in the pregnant endometrium, previous studies have demonstrated that ESFs retain a presence in the endometrium from the first trimester all the way to term (*Richards et al., 1995*; *Suryawanshi et al., 2018*; *Muñoz-Fernández et al., 2019*; *Sakabe et al., 2020*).

HAND2 protein was localized to nuclei in ESF lineage cells in human pregnant decidua (*Figure 2B*) from Human Protein Atlas IHC data (*Uhlén et al., 2015*). We also used existing functional genomics data to explore the regulatory status of the *HAND2* locus (see Materials and methods for references). Consistent with active expression, the *HAND2* locus in human DSCs is marked by histone modifications that typify enhancers (H3K27ac) and promoters (H3K4me3) and is located in a region of open chromatin as assessed by ATAC-, DNaseI- and FAIRE-Seq. Additionally, it is bound by transcription factors that establish endometrial stromal cell type identity and mediate decidualization, including the progesterone receptor (PGR), NR2F2 (COUP-TFII), GATA2, FOSL2, FOXO1, as well as polymerase II (*Figure 2C*). The *HAND2* promoter loops to several distal enhancers, as assessed by H3K27ac HiChIP data generated from a normal hTERT-immortalized endometrial cell line (E6E7hTERT), including a region bound by PGR, NR2F2, GATA2, FOSL2, and FOXO1, that also contains SNPs associated with gestation length in recent GWAS (*Warrington et al., 2019*; *Sakabe et al., 2020*; *Figure 2C*). *HAND2* was significantly upregulated by decidualization of human ESFs into DSCs by cAMP/progesterone treatment ($Log_2FC = 1.28$, $p = 2.62 \times 10^{-26}$, $FDR = 1.16 \times 10^{-24}$), and significantly downregulated by siRNAs targeting PGR ($Log_2FC = -0.90$, $p = 7.05 \times 10^{-15}$, $FDR = 2.03 \times 10^{-13}$) and GATA2 ($Log_2FC = -2.73$, $p = 0.01$, $FDR = 0.19$) (*Figure 2D*). In contrast, siRNA-mediated knockdown of neither NR2F2 ($Log_2FC = -0.91$, $p = 0.05$, $FDR = 1.0$) nor FOXO1 ($Log_2FC = 0.08$, $p = 0.49$, $FDR = 0.74$) significantly altered *HAND2* expression (*Figure 2D* and see Materials and methods for references).

## Differential *HAND2* expression throughout the menstrual cycle and pregnancy

Our observation that *HAND2* is progesterone responsive suggests it may be differentially expressed throughout the menstrual cycle and pregnancy. To explore this possibility, we utilized previously published gene expression datasets generated from the endometrium across the menstrual cycle (*Talbi et al., 2006*) and from the basal plate from mid-gestation to term (*Winn et al., 2007*). *HAND2* expression tended to increase from proliferative through the early and middle secretory phases, reaching a peak in the late secretory phase of the menstrual cycle (*Figure 2E*). In stark contrast, *HAND2* decreases in expression from the first trimester to term (*Figure 2F*). For comparison, 9% of genes were down-regulated between weeks 14–19 and 37–40 of pregnancy ($FDR \leq 0.10$). We also used previously published gene expression datasets to explore if *HAND2* was associated with disorders of pregnancy (*Lédée et al., 2011*; *Garrido-Gomez et al., 2017*). Although sample sizes of these datasets are small, and the intrinsic temporo-spatial heterogeneity of the endometrium remains a potential confounding factor, we found that *HAND2* was dysregulated in the endometria of women with implantation failure (IF) and recurrent spontaneous abortion (RSA), while it was not differentially expressed in ESFs or DSCs from women with preeclampsia (PE), compared to controls (*Figure 2G*).

## *HAND2* regulates a distinct set of target genes

*HAND2* expression has previously been shown to play a role in orchestrating the transcriptional response to progesterone during decidualization in human and mouse DSCs (*Huyen and Bany, 2011*; *Li et al., 2011*; *McConaha et al., 2011*; *Shindoh et al., 2014*; *Murata et al., 2020*). However,

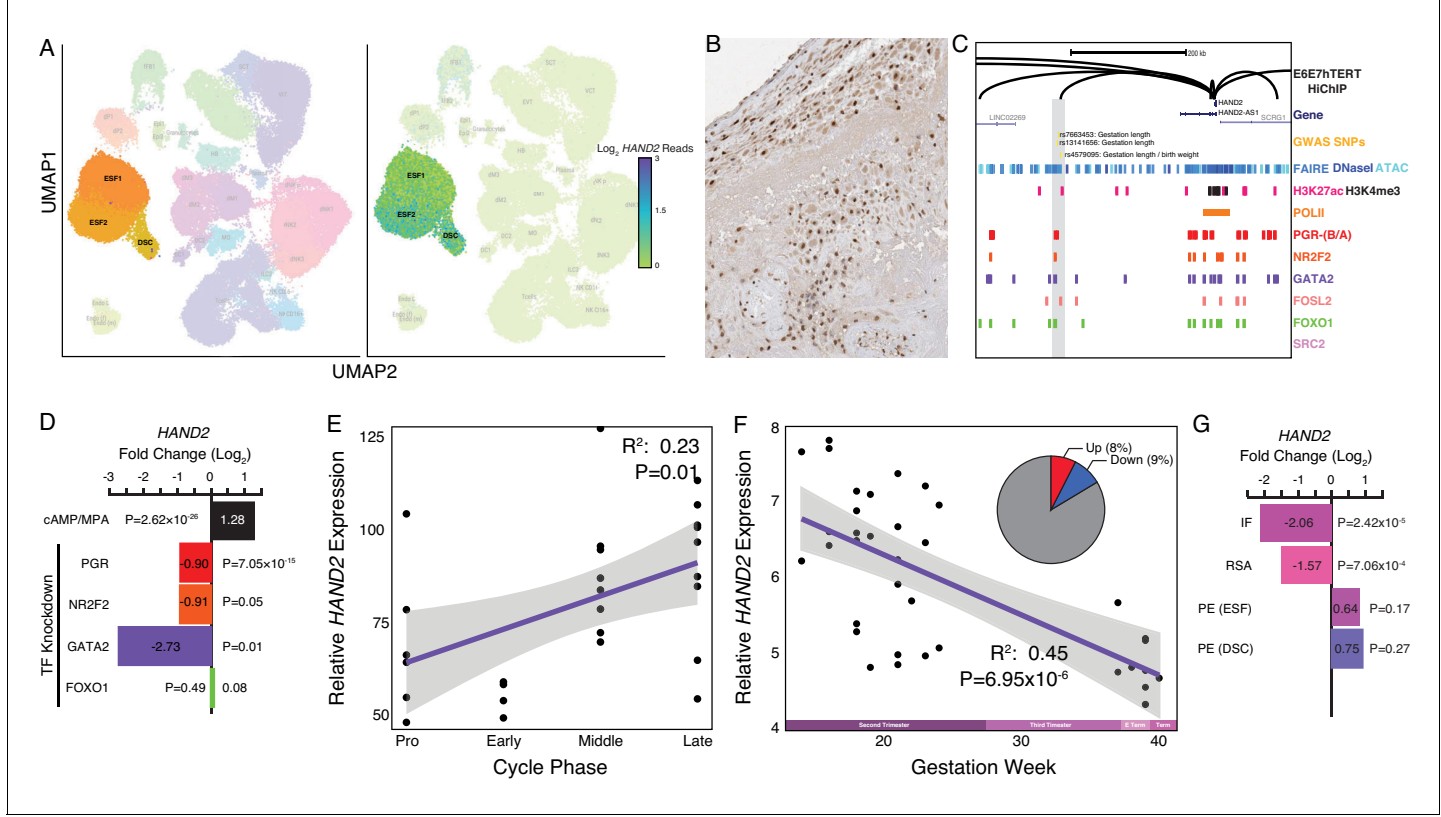

**Figure 2.** Expression of *HAND2* at the maternal-fetal interface. (**A**) UMAP clustering of scRNA-Seq data from human first trimester maternal-fetal interface. Left, clusters colored according to inferred cell type. The ESF1, ESF2, and DSC clusters are highlighted. Right, cells within clusters are colored according to *HAND2* expression level. scRNA-Seq data from *Vento-Tormo et al., 2018*. (**B**) HAND2 protein expression in human pregnant decidua, with strong staining and localization in the nuclei of endometrial stromal cells. Image credit: Human Protein Atlas. (**C**) Regulatory landscape of the *HAND2* locus. Chromatin loops inferred from H3K27ac HiChIP, regions of open chromatin inferred from FAIRE-, DNaseI, and ATAC-Seq, and the locations of histone modifications and transcription factor ChIP-Seq peaks are shown. The location of SNPs associated with gestation length / birth weight is also shown (highlighted in gray). Note that the *HAND2* promoter forms a long-range loop to a region marked by H3K27ac and bound by PGR, NR2F2 (COUP-TFII), GATA2, FOSL2, and FOXO1. (**D**) *HAND2* expression is up-regulated by *in vitro* decidualization of ESFs into DSC by cAMP/ progesterone treatment, and down-regulated by siRNA-mediated knockdown of PGR and GATA2, but not NR2F2 or FOXO1. n = 3 per transcription factor knockdown. (**E**) Relative expression of *HAND2* in the proliferative (n = 6), early (n = 4), middle (n = 9), and late (n = 8) secretory phases of the menstrual cycle. Note that outliers are excluded from the figure but not the regression; 95% CI is shown in gray. Gene expression data from *Talbi et al., 2006*. (**F**) Relative expression of *HAND2* in the basal plate from mid-gestation to term (14–40 weeks, n = 36); 95% CI is shown in gray. Inset, percent of up- (Up) and down-regulated (Down) genes between weeks 14–19 and 37–40 of pregnancy (FDR ≤ 0.10). Gene expression data from *Winn et al., 2007*. (**G**) *HAND2* expression is significantly down-regulated in the endometria of women with implantation failure (IF, n = 5) and recurrent spontaneous abortion (RSA, n = 5) compared to fertile controls (n = 5), but is not differentially expressed in ESFs or DSCs from women with preeclampsia (PE, n = 5) compared to healthy controls (n = 5). Gene expression data for RSA and IF from *Lédée et al., 2011* and for PE from *Garrido-Gomez et al., 2017*.

whether *HAND2* has functions in other endometrial stromal lineage cells such as ESFs, which persist in the pregnant endometrium till term (*Richards et al., 1995*; *Suryawanshi et al., 2018*; *Muñoz-Fernández et al., 2019*; *Sakabe et al., 2020*) but have received less attention than DSCs during pregnancy, is unknown. Therefore, we used siRNA to knockdown *HAND2* expression in human hTERT-immortalized ESFs (T-HESC) and assayed global gene expression changes by RNA-Seq 48 hr after knockdown. We found that *HAND2* was knocked down ~78% (p = $7.79 \times 10^{-3}$) by siRNA treatment (*Figure 3A*), which dysregulated the expression of 553 transcripts (489 genes) at FDR ≤ 0.10 (*Figure 3A* and *Figure 3—source data 1*). Genes dysregulated by *HAND2* knockdown were enriched in several pathways and human phenotype ontologies relevant to endometrial stromal cells and pregnancy in general (*Figure 3B* and *Figure 3—source data 2, 3*).

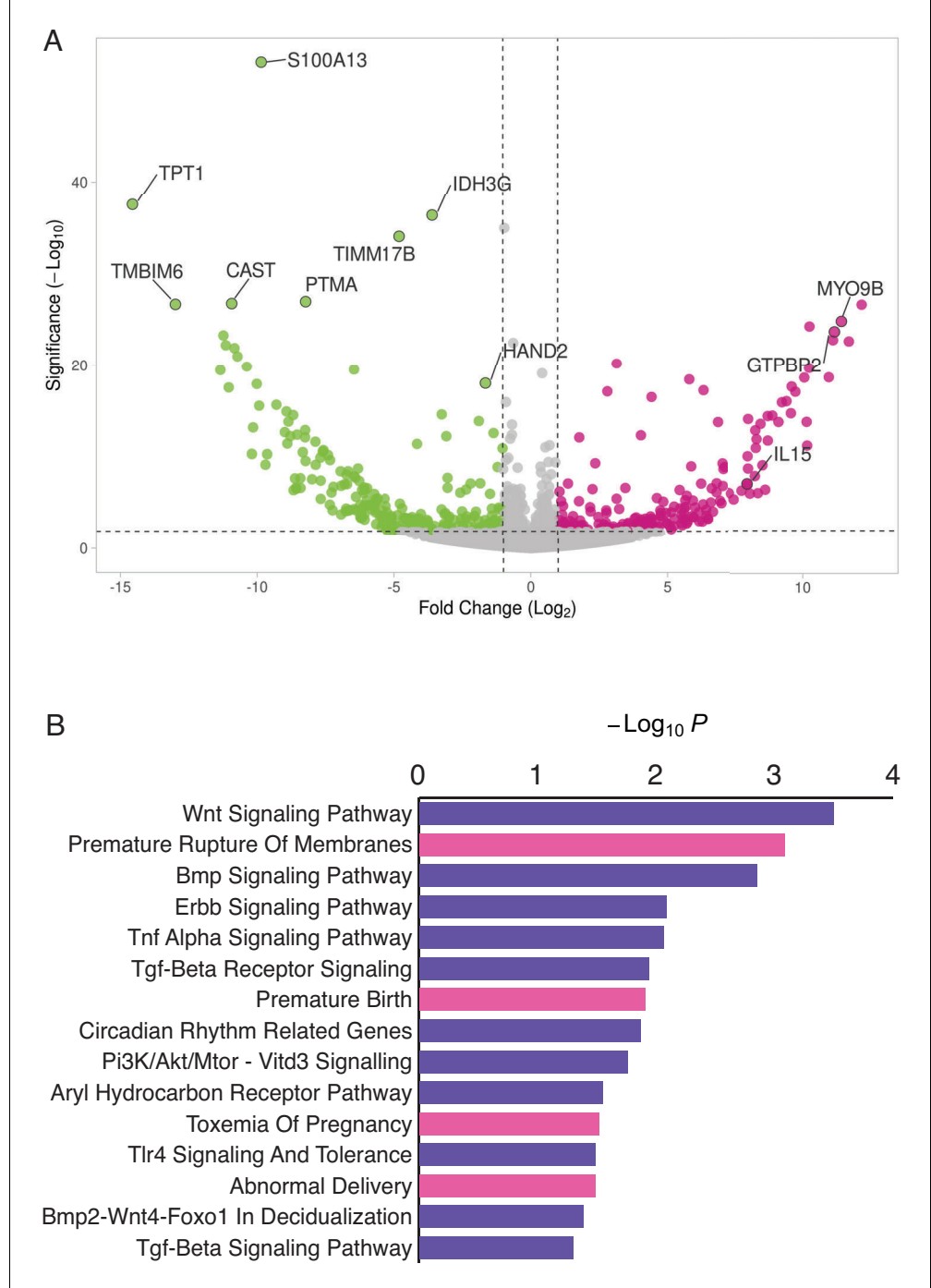

**Figure 3.** HAND2 regulates a distinct set of target genes, including *IL15*. (**A**) Volcano plot of gene expression upon *HAND2* knockdown. Only genes that are significantly differentially expressed (DE) with FDR ≤ 0.10 are colored. Genes with ≥ 1 fold changes in expression are shown in pink (up-regulated), green (down-regulated) or gray (not differentially expressed). X-axis shows $\log_2$ fold change, Y-axis shows Wald statistic p-value, horizontal dashed line indicates FDR = 0.10. Full list of DE genes can be found in *Figure 3—source data 1*. (**B**) Pathways (purple) and human phenotype ontologies (pink) in which genes dysregulated upon *HAND2* knockdown are enriched. We used a hypergeometric p-value to determine enriched pathway and disease ontology terms. The Benjamini-Hochburg Adjusted p-value (FDR), Odds Ratio, Combined Score, and Genes associated with each term can be found in *Figure 3—source data 2* and *Figure 3—source data 3*.

The online version of this article includes the following source data for figure 3:

**Source data 1.** Genes differentially expressed (DE) upon *HAND2* knockdown.

**Source data 2.** Pathways (Wikipathway 2019) enriched among genes differentially expressed by *HAND2* knockdown.

**Source data 3.** Human phenotype (disease) ontology terms enriched among genes differentially expressed by *HAND2* knockdown.

Enriched pathways play a role in decidualization (e.g. 'Wnt Signaling', 'BMP Signaling', 'ErbB Signaling', 'TGF-beta Receptor Signaling' and 'BMP2-WNT4-FOXO1 Pathway in Human Primary Endometrial Stromal Cell Differentiation'), as well as in placental bed development disorders and preeclampsia, the induction of pro-inflammatory factors via nuclear factor-κB (NFκB), mediation of maternal immunotolerance to the fetal allograft, circadian rhythm in association with implantation and parturition, and the decidual inflammation, senescence, and parturition. Selected pathways and associated references are listed in *Table 1*.

Enriched human phenotype ontology terms were related to complications of pregnancy, including 'Premature Rupture of Membranes', 'Premature Birth', 'Toxemia of Pregnancy' (preeclampsia) and 'Abnormal Delivery'. We also observed that several genes in the NFκB pathway, such as MYD88, CHUK, IκBKE, NFκBIE, and RTKN2 were differentially expressed; NFκB signaling has been associated with the molecular etiology of preterm birth (*Allport et al., 2001*; *Lindström and Bennett, 2005*).

## *HAND2* regulates *IL15* expression in endometrial stromal fibroblast lineage cells

Among the genes dysregulated by *HAND2* knockdown in ESFs was *IL15* (Log$_2$FC = 7.98, p = 7.91×10$^{-8}$, FDR = 1.49×10$^{-5}$), a pleiotropic cytokine previously shown to be expressed in the endometrium and decidua (*Figure 3A* and *Table 2*; *Kitaya et al., 2000*; *Okada et al., 2000*; *Dunn et al., 2002*; *Okada et al., 2004*; *Godbole and Modi, 2010*). *IL15* was robustly expressed at the first trimester maternal-fetal interface in stromal fibroblast lineage cells (*Figure 4A*), and there was a general correlation between *HAND2* and *IL15* expression in single cells (*Figure 4A* inset) (*Vento-Tormo et al., 2018*). IL15 protein localized to cytoplasm in ESF lineage cells in human pregnant decidua (*Figure 4B*) in Human Protein Atlas IHC data. The *IL15* promoter loops to several distal sites in H3K27ac HiChIP data from E6E7hTERT endometrial cells including to regions bound by PGR, NR2F2, GATA2, FOSL2, FOXO1, and SRC2, an intrinsic histone acetyltransferase that is a

**Table 1.** Genes dysregulated by *HAND2* knockdown are enriched in pathways relevant to endometrial stromal cells and pregnancy in general.

| Enriched pathway | Roles in ESFs and pregnancy | References |
|---|---|---|
| Wnt Signaling | Decidualization | *Peng et al., 2008*; *Hayashi et al., 2009*; *Sonderegger et al., 2010*; *Franco et al., 2011*; *Wang et al., 2013* |
| BMP Signaling | Decidualization | *Ying and Zhao, 2000*; *Lee et al., 2007*; *Li et al., 2007*; *Wetendorf and DeMayo, 2012* |
| ErbB Signaling | Decidualization | *Lim et al., 1997*; *Klonisch et al., 2001*; *Large et al., 2014* |
| TGF-beta Receptor Signaling | Decidualization | *Jones et al., 2006*; *Li, 2014*; *Ni and Li, 2017* |
| BMP2-WNT4-FOXO1 Pathway in Human Primary Endometrial Stromal Cell Differentiation | Decidualization | *Gellersen and Brosens, 2003*; *Buzzio et al., 2006*; *Lee et al., 2007*; *Li et al., 2007*; *Brayer et al., 2011*; *Lynch et al., 2009*; *Kajihara et al., 2013* |
| AGE/RAGE Pathway | Placental bed development disorders Preeclampsia Induction of pro-inflammatory factors via nuclear factor-κB (NFκB) | *Chekir et al., 2006*; *Lappas et al., 2007*; *Oliver et al., 2011*; *Guedes-Martins et al., 2013* |
| Aryl Hydrocarbon Receptor Pathway | Mediation of maternal immunotolerance to fetal allograft | *Munn et al., 1998*; *Abbott et al., 1999*; *Funeshima et al., 2005*; *Hao et al., 2013* |
| Circadian Rhythm Related Genes | Implantation and parturition | *Roizen et al., 2007*; *Olcese, 2012*; *Olcese et al., 2013*; *Greenhill, 2014*; *Menon et al., 2016* |
| RAC1/PAK1/p38/MMP2 Pathway | Decidual inflammation, senescence and parturition | *Menon et al., 2016* |

**Table 2.** Exemplar genes differentially expressed in ESFs upon siRNA-mediated *HAND2* knockdown.

Mean, base mean expression level. FC, log$_2$ fold change. SE, standard error in log$_2$ fold change. WS, Wald statistic. p-Value, Wald test p-value. Adj-p, Benjamini-Hochberg (BH) adjusted p-value. Function, function of gene inferred from Wikipathway 2019 human annotation. Association with preterm birth (PTB, HP:0001622) and premature rupture of membranes (PROM, HP:0001788) inferred from human phenotype ontology annotation*.

| Gene | Mean | FC | SE | WS | p-Value | Adj-p | Function |
|------|------|-----|-----|-----|---------|-------|----------|
| *ARNT2* | 44.47 | 1.20 | 0.28 | 4.30 | 1.73E-05 | 1.89E-03 | AHR signaling / Circadian rhythm |
| *ARNTL* | 42.52 | −2.95 | 0.80 | −3.67 | 2.46E-04 | 1.79E-02 | AHR signaling / Circadian rhythm |
| *IL15* | 63.31 | 7.98 | 1.49 | 5.37 | 7.91E-08 | 1.49E-05 | Cell migration |
| *BMP4* | 58.10 | −8.34 | 1.25 | −6.66 | 2.75E-11 | 8.28E-09 | Decidualization |
| *GSK3B* | 69.23 | −5.53 | 1.48 | −3.73 | 1.94E-04 | 1.49E-02 | Decidualization |
| *HAND2* | 136.06 | −1.63 | 0.18 | −8.85 | 9.10E-19 | 7.64E-16 | Knocked down gene |
| *CHUK* | 765.77 | 0.44 | 0.14 | 3.14 | 1.67E-03 | 7.94E-02 | NFκB pathway |
| *IκBKE* | 9.91 | −5.41 | 1.65 | −3.27 | 1.06E-03 | 5.72E-02 | NFκB pathway |
| *MYD88* | 95.02 | −0.93 | 0.26 | −3.55 | 3.88E-04 | 2.59E-02 | NFκB pathway |
| *NFκBIE* | 106.63 | −0.78 | 0.25 | −3.08 | 2.10E-03 | 9.53E-02 | NFκB pathway |
| *RTKN2* | 50.64 | 8.02 | 1.33 | 6.02 | 1.75E-09 | 4.05E-07 | NFκB pathway |
| *CRKL* | 1278.85 | −0.31 | 0.09 | −3.31 | 9.21E-04 | 5.10E-02 | PTB |
| *EOGT* | 233.65 | −2.77 | 0.80 | −3.45 | 5.63E-04 | 3.38E-02 | PTB |
| *LMNA* | 773.24 | 6.97 | 1.57 | 4.44 | 8.92E-06 | 1.06E-03 | PTB; PROM |
| *PEX11B* | 181.31 | 10.05 | 1.12 | 9.00 | 2.28E-19 | 2.06E-16 | PTB |
| *SERPINH1* | 1654.90 | −6.01 | 1.31 | −4.59 | 4.40E-06 | 5.61E-04 | PROM |
| *SLC17A5* | 826.28 | 0.76 | 0.13 | 5.81 | 6.34E-09 | 1.40E-06 | PTB |
| *ZMPSTE24* | 2140.67 | −0.19 | 0.05 | −4.10 | 4.10E-05 | 3.92E-03 | PTB; PROM |

*Extended list of genes associated with the pathologies of pregnancy and their expression levels can be found in **Table 2—source data 1** used to generate **Figures 2G** and **4G**. (GSE26787 = recurrent spontaneous abortion [RSA] and implantation failure [IF]; GSE91077 = ESFs and DSCs from women with preeclampsia [PE]).

The online version of this article includes the following source data for Table 2:

Source data 1. Differential expression of genes in RSA, IF and PE (in ESFs and DSCs) in relation to genes differentially expressed upon *HAND2* knockdown in ESFs.

transcriptional co-factor of ligand-dependent hormone receptors (*Figure 4C*; see Materials and methods for references). The *IL15* promoter also loops to a putative enhancer in its first intron that contains a PGR-binding site and SNPs marginally associated with a maternal effect on offspring birth weight (rs190663174, $p = 6 \times 10^{-4}$) by GWAS (*Warrington et al., 2019*). *IL15* was significantly upregulated by *in vitro* decidualization of human ESFs into DSCs by cAMP/progesterone treatment (Log$_2$FC = 2.15, $p = 2.58 \times 10^{-33}$, FDR = $1.59 \times 10^{-31}$), and significantly downregulated by siRNAs targeting PGR (Log$_2$FC = −1.24, $p = 6.23 \times 10^{-15}$, FDR = $1.80 \times 10^{-13}$) and GATA2 (Log$_2$FC = −2.08, $p = 4.16 \times 10^{-3}$, FDR = 0.14) (*Figure 4D*), but not NR2F2 (Log$_2$FC = 0.19, $p = 0.38$, FDR = 0.93) or FOXO1 (Log$_2$FC = 0.29, $p = 0.04$, FDR = 0.22) (*Figure 4D*; see Materials and methods for references). Although HAND2 binding data is not available for human stromal fibroblast lineage cells, several HAND2 binding motifs ($\geq 0.85$ motif match) are located within enhancers that loop to the *IL15* promoter.

## Differential *IL15* expression throughout the menstrual cycle and pregnancy

Our observations that *HAND2* is progesterone responsive and differentially expressed throughout the menstrual cycle and pregnancy suggest that *IL15*, which is controlled by HAND2 as well as cAMP/progesterone/PGR/GATA2, may be similarly regulated. Indeed, like *HAND2*, we found that *IL15* expression increased as the menstrual cycle progressed, peaking in the middle-late secretory phases (*Figure 4E*; *Talbi et al., 2006*) and decreased in expression from the first trimester to term

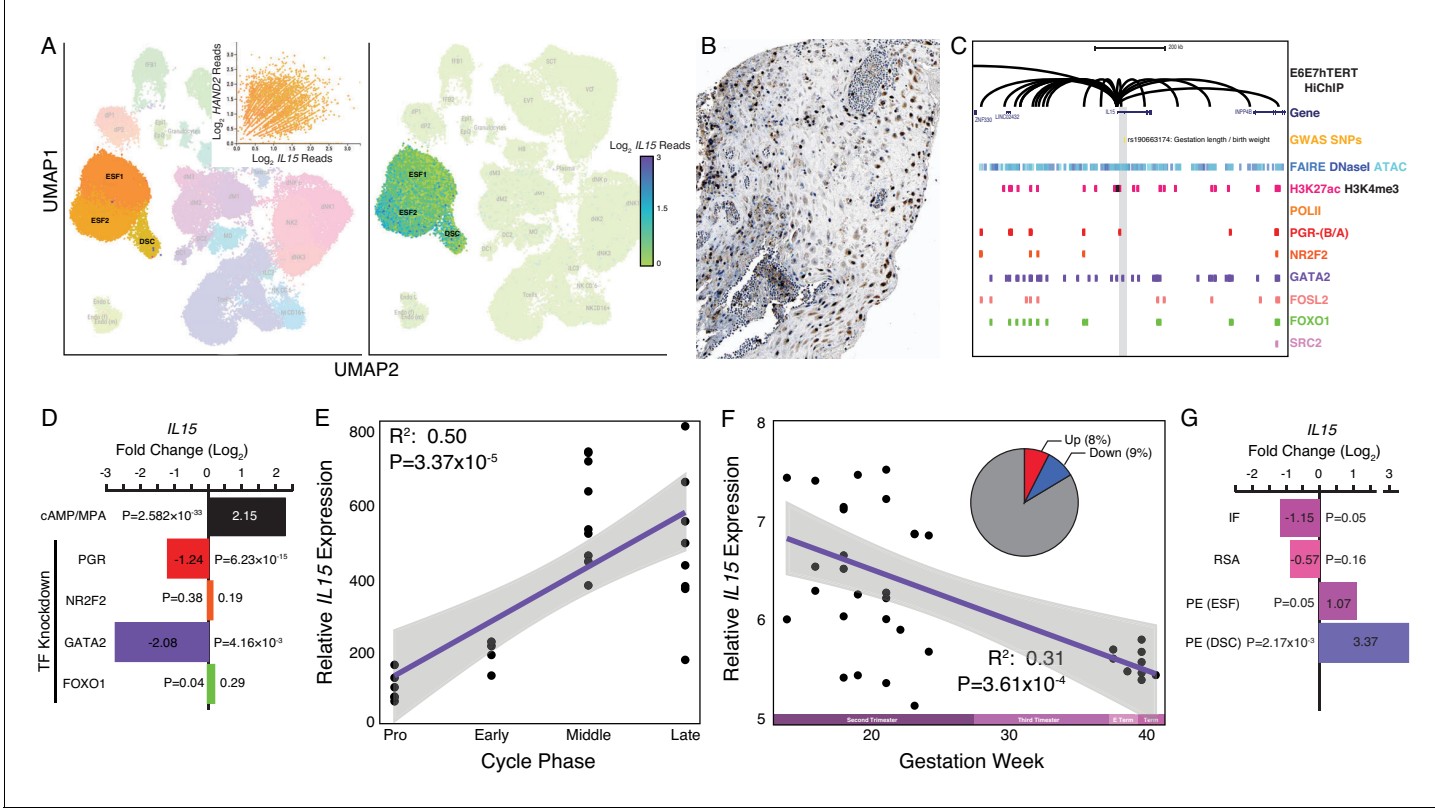

**Figure 4.** Expression of *IL15* at the maternal-fetal interface. (**A**) UMAP clustering of scRNA-Seq data from human first trimester maternal-fetal interface. Left, clusters colored according to inferred cell type. The ESF1, ESF2, and DSC clusters are highlighted. Inset, per cell expression of *HAND2* and *IL15* in ESF1s, ESF2s, and DSCs. Right, cells within clusters are colored according to *IL15* expression level. scRNA-Seq data from *Vento-Tormo et al., 2018*. (**B**) IL15 protein expression in human pregnant decidua, with strong cytoplasmic staining in endometrial stromal cells. Image credit: Human Protein Atlas. (**C**) Regulatory landscape of the *IL15* locus. Chromatin loops inferred from H3K27ac HiChIP, regions of open chromatin inferred from FAIRE-, DNaseI, and ATAC-Seq, and the locations of histone modifications and transcription factor ChIP-Seq peaks are shown. The location of an SNP associated with gestation length / birth weight is also shown (highlighted in gray). Note that the *IL15* promoter forms many long-range loops to regions marked by H3K27ac and bound by PGR, NR2F2 (COUP-TFII), GATA2, FOSL2, FOXO1, and SRC2. (**D**) *IL15* expression is upregulated by *in vitro* decidualization of ESFs into DSC by cAMP/progesterone treatment, and downregulated by siRNA-mediated knockdown of PGR and GATA2 but not NR2F2 or FOXO1. n = 3 per transcription factor knockdown. (**E**) Relative expression of *IL15* in the proliferative (n = 6), early (n = 4), middle (n = 9), and late (n = 8) secretory phases of the menstrual cycle. Note that outliers are excluded from the figure but not the regression; 95% CI is shown in gray. Gene expression data from *Talbi et al., 2006*. (**F**) Relative expression of *IL15* in the basal plate from mid-gestation to term (14–40 weeks, n = 36); 95% CI is shown in gray. Inset, percent of up- (Up) and downregulated (Down) genes between weeks 14–19 and 37–40 of pregnancy (FDR ≤ 0.10). Gene expression data from *Winn et al., 2007*. (**G**) *IL15* expression is significantly upregulated in DSCs from women with preeclampsia (PE, n = 5) compared to healthy controls (n = 5), while it is only marginally upregulated in ESFs from the same patient group. It is also marginally downregulated in the endometria of women with implantation failure (IF, n = 5) and it is not differentially expressed in the endometria of women with recurrent spontaneous abortion (RSA, n = 5) compared to fertile controls (n = 5). Gene expression data for RSA and IF from *Lédée et al., 2011* and for PE from *Garrido-Gomez et al., 2017*.

(*Figure 4F*; *Winn et al., 2007*). *IL15* expression was also dysregulated in the endometria of women with implantation failure but not recurrent spontaneous abortion, compared to fertile controls (*Figure 4G*; *Lédée et al., 2011*). In women with preeclampsia, *IL15* was not dysregulated in ESFs, but it was expressed significantly higher in DSCs, compared to controls (*Figure 4G*; *Garrido-Gomez et al., 2017*). Thus, like *HAND2*, *IL15* is differentially expressed throughout the menstrual cycle and pregnancy, and in the endometria of women with implantation failure.

## ESF-derived IL15 promotes NK and trophoblast migration

Endometrial stromal cells promote the migration of uterine natural killer (uNK) (*Chen et al., 2011*) and trophoblast cells (*Graham and Lala, 1991*; *Paiva et al., 2009*; *Zhu et al., 2009*; *Godbole et al., 2011*). IL15, in particular, stimulates the migration of uNK cells (*Allavena et al., 1997*; *Verma et al., 2000*; *Ashkar et al., 2003*; *Barber and Pollard, 2003*; *Kitaya et al., 2005*) and the human chorio-carcinoma cell line, JEG-3 (*Zygmunt et al., 1998*). Therefore, we tested whether ESF-derived IL15 influenced the migration of primary human NK cells and immortalized first trimester extravillous trophoblasts (HTR-8/SVneo) in trans-well migration assays (*Figure 5A*). We found that ESF media supplemented with recombinant human IL15 (rhIL15) was sufficient to stimulate the migration of NK and HTR-8/SVneo cells to the lower chamber of trans-wells, compared to non-supplemented media (*Figure 5B,C* and *Figure 5—source data 1*, *2*). Conditioned media from ESFs with siRNA-mediated *HAND2* knockdown increased migration of both NK and HTR-8/SVneo compared to negative control (i.e. non-targeting siRNA; *Figure 5B,C*). Conditioned media from ESFs with siRNA-mediated *IL15* knockdown reduced migration of both NK and HTR-8/SVneo cells compared to negative control (*Figure 5B,C*). Similarly, ESF conditioned media supplemented with anti-IL15 antibody reduced cell migration compared to media supplemented with control IgG antibody (*Figure 5B,C*).

## Discussion

Eutherian mammals evolved a suite of traits that support pregnancy, including an interrupted estrous cycle allowing for prolonged gestation lengths, maternal-fetal communication, implantation, maternal immunotolerance and recognition of pregnancy, and thus are uniquely afflicted by disorders of these processes. When searching for clues as to how variation in normal physiological functions can lead to dysfunction and disease, deeper understanding of the evolutionary and developmental histories of organ and tissues has the potential to provide novel insights. Here, we used evolutionary transcriptomics to identify genes that evolved to be expressed on the maternal side (endometrium) of the maternal-fetal interface during the origins of pregnancy in Eutherians, and hence may also contribute to pregnancy complications such as infertility, recurrent spontaneous abortion, and preterm birth.

Among the genes recruited into endometrial expression in Eutherians, we identified *HAND2*, a pleiotropic transcription factor that plays an essential role in suppressing estrogen signaling at the time of uterine receptivity to blastocyst embedding, through its down-regulation of pro-estrogenic genes and by directly inhibiting the transcriptional activities of the estrogen receptor (*Huyen and Bany, 2011*; *Li et al., 2011*; *Shindoh et al., 2014*; *Fukuda et al., 2015*; *Mestre-Citrinovitz et al., 2015*; *Murata et al., 2019*). Consistent with these functions, we found evidence of estrogen activity in the endometrium of pregnant opossum. Earlier research detected similar activity in the gravid oviduct of birds and reptiles (*Means et al., 1975*; *Kato et al., 1992*; *Girling, 2002*; *González-Morán, 2015*). These data indicate that suppression of estrogen signaling during the window of uterine receptivity to implantation is an evolutionary innovation of Eutherian mammals, which involved the recruitment of *HAND2* and its anti-estrogenic functions into endometrial expression.

The roles of *HAND2* in DSCs and implantation are well-understood (*Huyen and Bany, 2011*; *Li et al., 2011*; *Shindoh et al., 2014*; *Fukuda et al., 2015*; *Mestre-Citrinovitz et al., 2015*; *Murata et al., 2019*; *Šućurović et al., 2020*). In contrast, the function(s) of *HAND2* at other stages of pregnancy and in ESFs remain relatively unexplored, despite the persistence of ESFs in the pregnant endometrium until term (*Richards et al., 1995*; *Suryawanshi et al., 2018*; *Muñoz-Fernández et al., 2019*; *Sakabe et al., 2020*). *HAND2*, for example, plays a role in orchestrating the transcriptional response to progesterone during decidualization (*Huyen and Bany, 2011*; *Li et al., 2011*; *McConaha et al., 2011*; *Shindoh et al., 2014*; *Murata et al., 2020*). Similarly, *Hand2* knock-out mice are infertile because of persistent estrogen signaling during the window of implantation, leading to implantation failure (*Li et al., 2011*; *Jones et al., 2013*). The functions of *Hand2* at other stages of pregnancy could unfortunately not be explored in these *Hand2* knockout mice as the conditional targeting strategy knocks out *Hand2* in *Pgr*-expressing cells. *Hand2* expression is thereby eliminated upon initiation of *Pgr* expression in the uterus, coincident with the onset of sexual maturity.

We knocked down *HAND2* in ESFs and found that downstream dysregulated genes were enriched for human phenotype ontologies related to disorders of pregnancy, including 'Premature

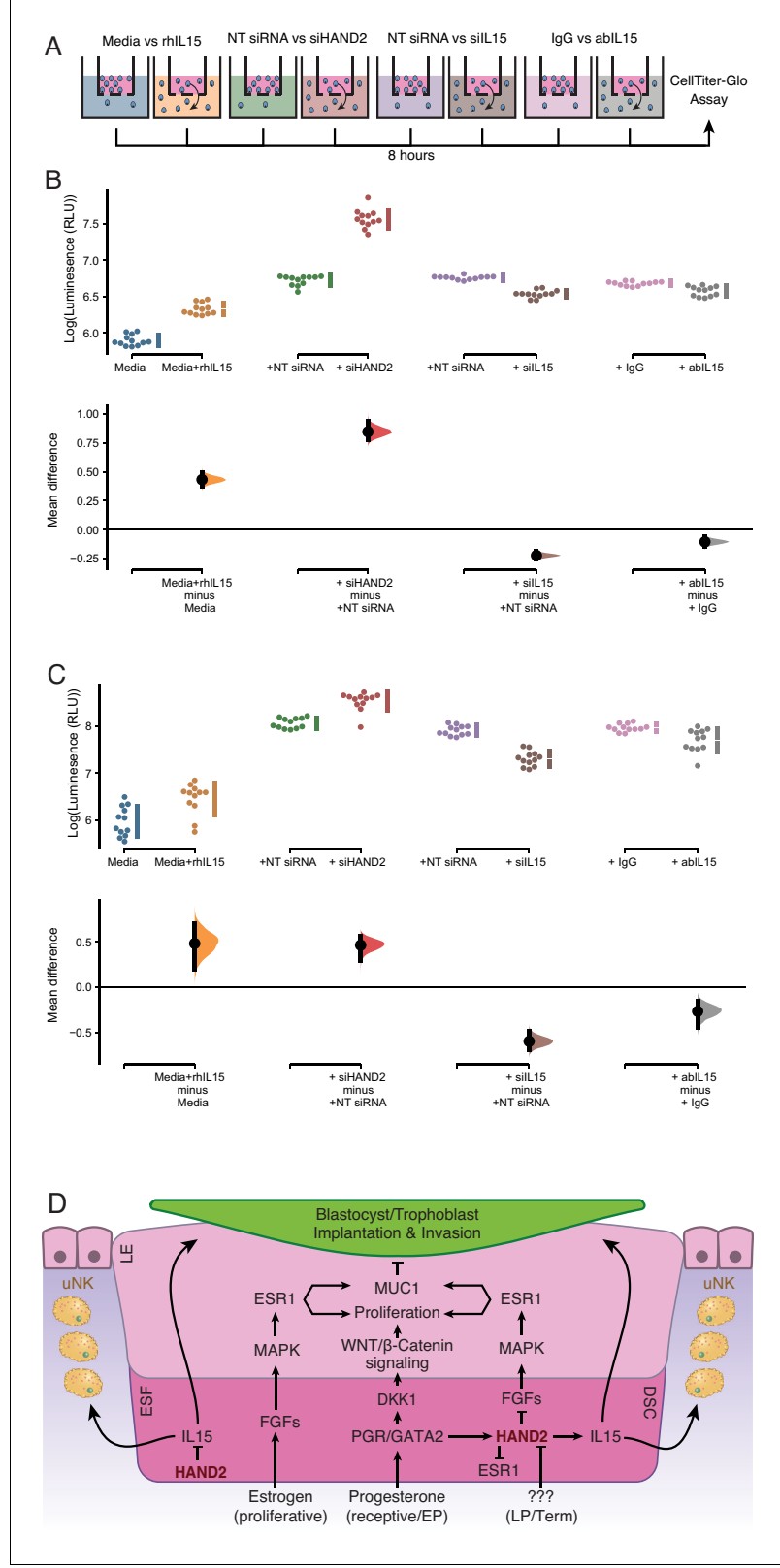

**Figure 5.** ESF-derived IL15 promotes NK and trophoblast migration in trans-well assays. (**A**) Cartoon of trans-well migration assay comparisons. Cells that migrated to the lower chamber were quantified using the CellTiter-Glo luminescent cell viability assay after 8 hr. (**B**) Primary natural killer (NK) cells. Raw luminescence data (RLU) from cells in the lower chamber is shown in the upper panel, mean difference (effect size) in experiment minus control

*Figure 5 continued on next page*

*Figure 5 continued*

luminescence values are shown as dots with the 95% confidence interval indicated by vertical bars in the lower panel; distribution estimated from 5000 bootstrap replicates. The mean difference between Media and media supplemented with recombinant human IL15 (Media+rhIL15) is 0.432 [95.0% CI: 0.376–0.492]; p = 0.00. The mean difference between ESFs transiently transfected with non-targeting siRNA (NT siRNA) and *HAND2*-specific siRNAs (siHAND2) is 0.847 [95.0% CI: 0.774–0.936]; p = 0.00. The mean difference between ESFs transiently transfected with NT siRNA and IL15-specific siRNAs (siIL15) is −0.223 [95.0% CI: −0.256 − −0.192]; p = 0.00. The mean difference between ESF media neutralized with a non-specific antibody (IgG) or IL15-specific antibody (abIL15) is −0.106 [95.0% CI: −0.147 − −0.067]; p = 0.00. n = 12. (**C**) Extravillous trophoblast cell line HTR-8/SVneo. Raw luminescence data (RLU) from cells in the lower chamber is shown in the upper panel, mean difference (effect size) in experiment minus control luminescence values are shown as dots with the 95% confidence interval indicated by vertical bars in the lower panel; distribution estimated from 5000 bootstrap replicates. The mean difference between Media and Media+rhIL15 is 0.482 [95.0% CI: 0.193–0.701]; p = 0.002. The mean difference between ESFs transiently transfected with NT siRNA and siHAND2 is 0.463 [95.0% CI: 0.291–0.559]; p = 0.00. The mean difference between ESFs transiently transfected with NT siRNA and siIL15 is −0.598 [95.0% CI: −0.698–0.490]; p = 0.00. The mean difference between ESF media neutralized with IgG or abIL15 is −0.267 [95.0% CI: −0.442 − −0.151]; p = 0.0004. n = 12. (**D**) Model of HAND2 functions in the endometrium. During the proliferative phase HAND2 inhibits *IL15*, and thus the migration of uNK and trophoblast cells. In the receptive phase, HAND2 activates *IL15*, which promotes migration of uNK and trophoblast cells. In the receptive phase and early pregnancy (EP), HAND2 suppresses estrogen signaling by down-regulating *FGF*s and directly binding and inhibiting the ligand-dependent transcriptional activation function of ESR1. During late pregnancy/term (LP/Term), reduced HAND2 expression mitigates its anti-estrogenic functions. Parturition signal unknown (???). ESF = endometrial stromal fibroblasts (proliferative phase), DSC = decidual stromal cells (receptive phase), LE = luminal epithelium.

The online version of this article includes the following source data for figure 5:

**Source data 1.** Raw and log transformed luminescence data for NK trans-well migration assays.

**Source data 2.** Raw and log transformed luminescence data for HTR-8 trans-well migration assays.

---

Rupture of Membranes', 'Premature Birth', and 'Abnormal Delivery', suggesting that *HAND2* has functions throughout pregnancy and in parturition. Indeed, we discovered that SNPs recently implicated in the regulation of gestation length and birth weight by GWAS (*Warrington et al., 2019*; *Sakabe et al., 2020*) make long-range interactions to the *HAND2* promoter. Also of note, *HAND2* expression is significantly higher in placental villous samples from idiopathic spontaneous preterm birth (isPTB) compared to term controls (*Brockway et al., 2019*). However, this difference may be related to gestational age rather than the etiology of PTB (*Eidem et al., 2016*).

Additionally, analyzing previously published datasets we noticed that *HAND2* expression decreases throughout gestation. Unlike the majority of Eutherians, where parturition closely follows the significant drop in progesterone concentrations in maternal peripheral blood, this is not the case for humans and other Old World primates (*Ratajczak et al., 2010*), where progesterone levels keep rising throughout gestation, reaching maximum at birth. These observations, combined with the central role of *HAND2* in mediating the anti-estrogenic actions of progesterone, suggest decreased *HAND2* at the end of pregnancy may contribute to the estrogen-dominant uterine environment at the onset of labor (*Pinto et al., 1966*; *Pepe and Albrecht, 1995*; *Mesiano et al., 2002*; *Smith et al., 2009a*; *Ratajczak et al., 2010*; *Welsh et al., 2012*), despite high systemic progesterone. Low *HAND2* at the end of pregnancy in humans is therefore most likely not directly related to progesterone concentrations, suggesting that an unidentified inhibitory signal reduces endometrial *HAND2* expression. While taken collectively these data indicate a role for *HAND2* in pre/term birth, a direct mechanistic link between *HAND2* expression and the timing of parturition remains to be demonstrated.

One of the genes dysregulated by *HAND2* knockdown was the multifunctional cytokine *IL15*, which plays important roles in innate and adaptive immunity. In the context of pregnancy, it is important for the recruitment of uterine natural killer (uNK) cells to the endometrium (*Kitaya et al., 2000*; *Verma et al., 2000*; *Ashkar et al., 2003*; *Barber and Pollard, 2003*; *Kitaya et al., 2005*; *Laskarin et al., 2006*). The roles of uNK cells in the remodeling of uterine spiral arteries and regulating trophoblast invasion are well known (*Zygmunt et al., 1998*; *Hanna et al., 2006*; *Smith et al., 2009b*; *Burke et al., 2010*; *Hazan et al., 2010*; *Lash et al., 2010*; *Bany et al., 2012*; *Robson et al.,*

*2012*; *Zhang et al., 2013*; *Lima et al., 2014*; *Fraser et al., 2015*; *Felker and Croy, 2016*; *Renaud et al., 2017*). Endometrial stromal cell-derived IL15 is also necessary for the selective targeting and clearance of senescent endometrial stromal cells from the implantation site by uNK cells, which is essential for endometrial rejuvenation and remodeling at embryo implantation (*Brighton et al., 2017*). Dysregulation of uNK cell-mediated clearance of these senescent cells has also been implicated in recurrent pregnancy loss (*Lucas et al., 2020*). We found that, like *HAND2*, *IL15* decreases throughout gestation and both genes increase in expression as the menstrual cycle progresses.

Unexpectedly, however, while previous studies showed that *HAND2* induces *IL15* in DSCs (*Shindoh et al., 2014*; *Murata et al., 2020*), we discovered that *HAND2* inhibited *IL15* expression in ESFs, indicating a switch in regulatory activity sometime during early menstrual cycle. These data suggest that *HAND2* regulates the appropriate timing of endometrial *IL15* expression during the menstrual cycle and throughout pregnancy, and thus the appropriate timing of uNK cell recruitment, trophoblast migration and the clearance of senescent endometrial stromal cells from the implantation site (*Figure 5D*). While the signals that initiate parturition in humans and other Catarrhine primates are unknown, it has been proposed that a 'decidual clock' may regulate the successful establishment and maintenance of pregnancy, such that severe decidualization defects lead to infertility, moderate defects lead to recurrent pregnancy loss/recurrent spontaneous abortion, and mild defects lead to preterm birth (*Norwitz et al., 2015*). Our results suggest that part of this clock may be the transition of ESFs that persist in the endometrium during pregnancy to DSCs, and/or DSCs into senescent DSCs (snDSCs) which no longer express *IL15* (*Lucas et al., 2020*), leading to a shift in the balance of tolerizing immune cells at the maternal-fetal interface and thus withdrawal of maternal immunotolerance of the fetal allograft. Thus, our observation of low *HAND2* and *IL15* near term may reflect a reduction of anti-inflammatory DSCs (high *HAND2* and *IL15*) and an accumulation of pro-inflammatory snDSCs (low *HAND2* and *IL15*) at the maternal-fetal interface. Additional work is needed to elucidate the mechanisms that underlie change in *HAND2-IL15* dynamics and determine whether these progressive cell state changes occur during gestation.

Decreased *HAND2* and *IL15* expression near term and their influence on immune cells at the maternal-fetal interface may also play a role in parturition. Pre/term labor is known to be associated with elevated inflammation and an influx of immune cells into utero-placental tissues (*Thomson et al., 1999*; *Young et al., 2002*; *Osman et al., 2003*; *Gomez-Lopez et al., 2010*; *Rinaldi et al., 2011*; *Rinaldi et al., 2015*; *Hamilton et al., 2012*; *Shynlova et al., 2013*; *Bartmann et al., 2014*; *Menon et al., 2016*; *Peters et al., 2016*; *Wilson and Mesiano, 2020*). uNK cells are abundant throughout gestation (*Bulmer et al., 1991*; *King et al., 1991*; *Moffett-King, 2002*; *Williams et al., 2009*; *Bartmann et al., 2014*), but whether they play a role in late pregnancy and parturition is unclear. However, depletion of uNK cells rescues LPS-induced preterm birth in *IL10*-null mice (*Murphy et al., 2005*), indicating they contribute to infection/inflammation-induced preterm parturition (*Murphy et al., 2009*). CD16$^+$CD56$^{dim}$ (cytotoxic) uNK cells have also been observed in the decidua and the placental villi of women with preterm but not term labor, suggesting an association between dysregulation of uNK cells and preterm birth in humans (*Gomaa et al., 2017*). uNK cells are associated with other pregnancy complications in humans such as fetal growth restriction, preeclampsia, and recurrent spontaneous abortion (*Moffett et al., 2004*; *Hiby et al., 2010*; *Wallace et al., 2013*; *Wallace et al., 2014*; *Kieckbusch et al., 2014*). Taken together, these data indicate that uNK cells may act downstream of *HAND2-IL15* signaling in the timing of parturition.

## Conclusions

Here, we show that *HAND2* evolved to be expressed in endometrial stromal cells in the Eutherian stem-lineage, coincident with the evolution of suppressed estrogen signaling during the window of implantation and an interrupted reproductive cycle during pregnancy, which necessitated a means to regulate the length of gestation. Our data suggest that *HAND2* may contribute to the regulation of gestation length by promoting an estrogen dominant uterine environment near term and through its effect on IL15 signaling and uNK cell function. To further expand our understanding of *HAND2* functions at the molecular mechanistic level, multiple technical and ethical difficulties associated with studying human pregnancy *in vivo* will need to be overcome. Therefore, recently developed organoid models of the human maternal-fetal interface (*Rinehart et al., 1988*; *Boretto et al., 2017*;

*Turco et al., 2017*; *Turco et al., 2018*; *Marinić et al., 2020*), which allow for *in vitro* 3D manipulation, will be instrumental.

## Materials and methods

### Endometrial gene expression profiling and ancestral transcriptome reconstruction

#### Data collection
We obtained previously generated RNA-Seq data from endometria of amniotes by searching NCBI BioSample, Sequence Read Archive (SRA), and Gene Expression Omnibus (GEO) databases for anatomical terms referring to the portion of the female reproductive tract (FRT), including 'uterus', 'endometrium', 'decidua', 'oviduct', and 'shell gland', followed by manual curation to identify those datasets that included the FRT region specialized for maternal-fetal interaction or shell formation. Datasets that did not indicate whether samples were from pregnant or gravid females were excluded, as were those composed of multiple tissue types. Species included in this study and their associated RNA-Seq accession numbers are included in *Figure 1—source data 1*.

#### New RNA-Seq data
Endometrial tissue samples from the pregnant uteri of baboon (n = 3), mouse (n = 3), hamster (n = 3), bat (n = 2), and squirrel (n = 2) were dissected and mailed to the University of Chicago in RNA-Later. These samples were further dissected to remove myometrium, luminal epithelium, and extra-embryonic tissues, and then washed three times in ice cold PBS to remove unattached cell debris and red blood cells. Total RNA was extracted from the remaining tissue using the RNeasy Plus Mini Kit (74134, QIAGEN) per manufacturer's instructions. RNA concentrations were determined by Nanodrop 2000 (Thermo Scientific). A total amount of 2.5 µg of total RNA per sample was submitted to the University of Chicago Genomics Facility for Illumina Next Gen RNA sequencing. Quality was assessed with the Bioanalyzer 2100 (Agilent). A total RNA library was generated using the TruSEQ stranded mRNA with RiboZero depletion (Illumina) for each sample. The samples were fitted with one of six different adapters with a different 6-base barcode for multiplexing. Completed libraries were run on an Illumina HiSEQ2500 with v4 chemistry on two replicate lanes for hamster and one lane for other species of an eight lane flow cell, generating 30–50 million 50 bp single-end reads per sample.

#### Multispecies RNA-Seq analysis
For all RNA-Seq analyses, we used Kallisto (*Bray et al., 2016*) version 0.42.4 to pseudo-align the raw RNA-Seq reads to reference transcriptomes (see *Figure 1—source data 1* for reference genome assemblies) and to generate transcript abundance estimates. We used default parameters bias correction, and 100 bootstrap replicates. Kallisto outputs consist of transcript abundance estimates in Transcripts Per Million (TPM), which were used to determine gene expression.

#### Ancestral transcriptome reconstruction
We previously showed that genes with TPM $\geq$ 2.0 are actively transcribed in endometrium while genes with TPM < 2.0 lack hallmarks of active transcription such as promoters marked with H3K4me3 (*Wagner et al., 2012*; *Wagner et al., 2013*). Based on these findings, we transformed values of transcript abundance estimates into discrete character states, such that genes with TPM $\geq$ 2.0 were coded as expressed (state = 1), genes with TPM < 2.0 were coded as not expressed (state = 0), and genes without data in specific species coded as missing (state = ?). The binary encoded endometrial gene expression dataset generally grouped species by phylogenetic relatedness, suggesting greater signal-to-noise ratio than raw transcript abundance estimates. Therefore, we used the binary encoded endometrial transcriptome dataset to reconstruct ancestral gene expression states and trace the evolution of gene expression gains (0 → 1) and losses (1 → 0) in the endometria across vertebrate phylogeny (*Figure 1A*). We used Mesquite (*Maddison and Maddison, 2019*) (v3.02) with parsimony optimization to reconstruct ancestral gene expression states, and identify genes that gained or lost endometrial expression. Expression was classified as an unambiguous gain if a gene

was not inferred as expressed at a particular ancestral node (state = 0) but inferred as expressed (state = 1) in a descendent of that node, and vice versa for the classification of a loss of endometrial expression (*Figure 1—source data 2*). Parsimony optimization of ancestral states results in three ancestral state reconstructions (ASRs): 'ambiguous', 'most-parsimonious', and 'unambiguous'. An ambiguous ASR is not resolved and interpreted as unknown, a most-parsimonious ASR is 'potentially' the ASR but not necessarily so because other ancestral states are also possible, while an unambiguous ASR is the most-optimal state such that alternative ASRs can be rejected. The criterion for determining which genes belong in these categories was the parsimony optimization method implemented in Mesquite (v3.02). We thus identified 149 genes that unambiguously evolved endometrial expression in the stem-lineage of Eutherian mammals (*Figure 1—source data 3*).

### Pathway enrichments

We used WebGestalt v. 2019 (*Liao et al., 2019*) to determine if the 149 identifies genes were enriched in ontology terms using over-representation analysis (ORA). A key advantage of WebGestalt is that it allowed for the inclusion of a custom background gene list, which was the set of 21,750 genes for which we could reconstruct ancestral states, rather than all annotated protein-coding genes in the human genome. We used ORA to identify enriched terms for three pathway databases (KEGG, Reactome, and Wikipathway), the Human Phenotype Ontology database, and a custom database of genes implicated in preterm birth by GWAS. The preterm birth gene set was assembled from the NHGRI-EBI Catalog of published genome-wide association studies (GWAS Catalog), including genes implicated in GWAS with either the ontology terms 'Preterm Birth' (EFO_0003917) or 'Spontaneous Preterm Birth' (EFO_0006917), as well as two recent preterm birth GWAS (*Warrington et al., 2019*; *Sakabe et al., 2020*) using a genome-wide significant p-value of $9 \times 10^{-6}$. The custom gmt file used to test for enrichment of preterm birth associated genes is included as a supplementary data file to *Figure 1* (*Figure 1—source data 8*).

## Marsupial gene expression analysis

### Opossum uterine gene expression time course

To explore the expression of *HAND2* and other genes throughout gestation in Marsupials, we analyzed previously generated short-tailed opossum (*Monodelphis domestica*) RNA-Seq data from virgin non-pregnant (SRR2972837, SRR2972848), day 7 (during the histotrophic phase; SRP111668), day 12.5 (just after hatching and during the transition from the histotrophic to the placental phase; GSM1611397), day 13 (early placental phase; SRP111668), day 13.5–14.5 (during the late placental to early parturition phase; SRR2969483, SRR2969536, SRR2970443), and 9–10 month postpartum utera (SRR2972728 and SRR2972729) (*Figure 1—source data 7*; *Lynch et al., 2015*; *Hansen et al., 2016*; *Griffith et al., 2017*; *Griffith et al., 2019*).

### Wallaby RNA-Seq data from non-pregnant and pregnant animals

RNA-Seq data for tammar wallaby (*Macropus eugenii*) uterine tissue from pregnant and non-pregnant animals were from PRJDB1934.

### RNA-Seq analysis

We used Kallisto version 0.42.4 to pseudo-align RNA-Seq reads to the *M. domestica* and *M. eugenii* reference transcriptomes with default parameters bias correction and 100 bootstrap replicates. Kallisto output quantifies transcript abundance estimates in TPM.

## Immunohistochemistry (IHC)

Endometrial tissue from pregnant opossum (12.5d) was fixed in 10% neutral-buffered formalin, paraffin-embedded, sectioned at 4 µm, and mounted on slides. Paraffin sections were dried at room temperature overnight and then baked for 12 hr at 50°C. Prior to immunostaining, de-paraffinization and hydration were done in xylene and graded ethanol to distilled water. During hydration, a 5 min blocking for endogenous peroxidase was done in 0.3% $H_2O_2$ in 95% ethanol. Antigen retrieval was performed in retrieval buffer pH 6, using a pressure boiler microwave as a heat source with power set to full, allowing retrieval buffer to boil for 20 min, and then cooled in a cold water bath for 10 min. To stain sections, we used the Pierce Peroxidase IHC Detection Kit (cat # 36000) following

the manufacturer's protocol. Briefly, uterine sections were incubated at 4°C overnight with polyclonal antibodies against HAND2 (Santa Cruz SC-9409), MUC1 (Novus Biologicals NB120-15481), p-ERα (Santa Cruz SC-12915), p-Erk1/2 (also known as MAPK1/2; Santa Cruz SC-23759-R) at 1:1000 dilution in blocking buffer. The next day sections were washed 3x in wash buffer, and incubated with HRP-conjugated rabbit anti-mouse IgG (H+L) secondary antibody (Invitrogen cat # 31450) at 1:10,000 dilution in blocking buffer. After 30 min at 4°C, slides were washed 3x in wash buffer. Slides were developed with 1x DAB/metal concentrate and stable peroxide buffer for 5 min, then rinsed 3x for 3 min in wash buffer, and mounted with Permount (SP15-100; Thermo Fisher Scientific).

## Expression of *HAND2* and *IL15* at the maternal-fetal interface

We used previously published single-cell RNA-Seq (scRNA-Seq) data from the human first trimester maternal-fetal interface (*Vento-Tormo et al., 2018*) to determine which cell types express *HAND2* and *IL15*. The dataset consists of transcriptomes for ~70,000,000 individual cells of many different cell types, including: three populations of tissue resident decidual natural killer cells (dNK1, dNK2, and dNK3), a population of proliferating natural killer cells (dNKp), type two and/or type three innate lymphoid cells (ILC2/ILC3), three populations of decidual macrophages (dM1, dM2, and dM3), two populations of dendritic cells (DC1 and DC2), granulocytes (Gran), T cells (TCells), maternal and lymphatic endothelial cells (Endo), two populations of epithelial glandular cells (Epi1 and Epi2), two populations of perivascular cells (PV1 and PV2), two endometrial stromal fibroblast populations (ESF1 and ESF2), and decidual stromal cells (DSCs), placental fibroblasts (fFB1), extravillous- (EVT), syncytio- (SCT), and villus- (VCT) cytotrophoblasts (*Figure 2A*). Data were not reanalyzed, rather previously analyzed data were accessed using the cell×gene website available at https://maternal-fetal-interface.cellgeni.sanger.ac.uk.

We note that Vento-Tormo et al. identified five populations of cells in the endometrial stromal lineage, including two perivascular populations (likely reflecting the mesenchymal stem cell-like progenitor of endometrial stromal fibroblasts and decidual stromal cells) and three cell types they call 'decidual stromal cells' and label 'dS1-3'. However, based on the gene expression patterns of 'dS1-3' (shown in Vento-Tormo et al. Figure 3a), only 'dS3' are decidualized, as indicated by expression of classical markers of decidualization such and *PRL* (*Tabanelli et al., 1992*) and *IGFBP1/2/6* (*Tabanelli et al., 1992*; *Kim et al., 1999*). In stark contrast, 'dS1' do not express decidualization markers but highly express markers of ESFs such as *TAGLN* and *ID2*, as well as markers of proliferating ESFs including *ACTA2* (*Kim et al., 1999*). 'dS2' also express ESFs markers (*TAGLN*, *ID2*, *ACTA2*), but additionally *LEFTY2* and *IGFBP1/2/6*, consistent with ESFs that have initiated the process of decidualization. These data indicate that the 'dS1' and 'dS2' populations are both ESFs, but 'dS2' are ESFs that have initiated decidualization (because they express *IGFBPs* but not *PRL*), and that 'dS3' are DSCs. Vento-Tormo et al. show that the differences in gene expression between 'dS1-3' are related to their topography in the endometrium, but degree of decidualization ('dS1'/ESF1 < 'dS2'/ESF2 < 'dS3'/DSC) is also linked to differential gene expression.

Consistent with this, other scRNA-Seq studies have identified two ESF populations and one DSC population in the first trimester decidua, and used pseudotime analyses to show that they represent different states of differentiation from ESFs to mature DSCs (*Suryawanshi et al., 2018*). Therefore, we prefer to use the ESF1/ESF2/DSC nomenclature because it more accurately reflects the biology and gene expression profile of these cell types than the 'dS1-3' naming convention. We also note that while it is generally thought that ESFs are absent from the pregnant uterus, ESFs retain a presence in the endometrium from the first trimester until term (*Richards et al., 1995*; *Suryawanshi et al., 2018*; *Muñoz-Fernández et al., 2019*; *Sakabe et al., 2020*).

## Expression of HAND2 and IL15 in human decidual cells

We used previously published IHC data for HAND2 and IL15 generated from pregnant human decidua as part of the Human Protein Atlas project (http://www.proteinatlas.org/; *Uhlén et al., 2015*). Image/gene/data available from IL15 (https://www.proteinatlas.org/ENSG00000164136-IL15/tissue) and HAND2 (https://www.proteinatlas.org/ENSG00000164107-HAND2/tissue).

## Functional genomic analyses of the *HAND2* and *IL15* loci

### Gene expression data

We used previously published RNA-Seq and microarray gene expression data generated from human ESFs and DSCs that were downloaded from National Center for Biotechnology Information (NCBI) Sequence Read Archive (SRA) and processed remotely using Galaxy platform (https://usegalaxy.org/; Version 20.01) (*Afgan et al., 2018*) for RNA-Seq data and GEO2R. RNA-Seq datasets were transferred from SRA to Galaxy using the Download and Extract Reads in FASTA/Q format from NCBI SRA tool (version 2.10.4+galaxy1). We used HISAT2 (version 2.1.0+galaxy5; *Kim et al., 2015*) to align reads to the Human hg38 reference genome using single- or paired-end options depending on the dataset and unstranded reads, and report alignments tailored for transcript assemblers including StringTie. Transcripts were assembled and quantified using StringTie (v1.3.6) (*Pertea et al., 2015*; *Pertea et al., 2016*), with reference file to guide assembly and the 'reference transcripts only' option, and output count files for differential expression with DESeq2/edgeR/limma-voom. Differentially expressed genes were identified using DESeq2 (version 2.11.40.6+galaxy1) (*Anders and Huber, 2010*; *Love et al., 2014*). The reference file for StringTie guided assembly was wgEncodeGencodeBasicV33. GEO2R performs comparisons on original submitter-supplied processed data tables using the GEOquery (*Davis and Meltzer, 2007*) and limma (*Smyth et al., 2002*) R packages from the Bioconductor project (https://bioconductor.org/; *Gentleman et al., 2004*).

Datasets included gene expression profiles of primary human ESFs treated for 48 hr with control non-targeting, PGR-targeting (GSE94036), FOXO1-targeting (GSE94036) or NR2F2 (COUP-TFII)-targeting (GSE47052) siRNA prior to decidualization stimulus for 72 hr; transfection with GATA2-targeting siRNA was followed immediately by decidualization stimulus (GSE108407). We also explored the expression of *HAND2* and *IL5* in the endometria of women with recurrent spontaneous abortion and implantation failure using a previously published dataset (GSE26787), as well as in the endometrium throughout the menstrual cycle (GSE4888) and basal plate throughout gestation (GSE5999); HAND2 probe 220480_at, IL15 probe 205992_s_at. The expression of *HAND2* and *IL5* in ESFs and DSCs from women with normal pregnancy and severe preeclampsia was assessed from a previously generated Agilent Whole Human Genome Microarray 4 × 44K v2 dataset (GSE91077).

### ChIP-Seq and open chromatin data

We used previously published ChIP-Seq data generated from human DSCs that were downloaded from NCBI SRA and processed remotely using Galaxy (*Afgan et al., 2018*). ChIP-Seq reads were mapped to the human genome (GRCh37/hg19) using HISAT2 (*Kim et al., 2015*) with default parameters and peaks called with MACS2 (*Zhang et al., 2008*; *Feng et al., 2012*) with default parameters. Samples included PLZF (GSE75115), H3K4me3 (GSE61793), H3K27ac (GSE61793), H3K4me1 (GSE57007), PGR (GSE69539), the PGR A and B isoforms (GSE62475), NR2F2 (GSE52008), FOSL2 (GSE69539), FOXO1 (GSE69542), PolII (GSE69542), GATA2 (GSE108408), SRC-2/NCOA2 (GSE123246), AHR (GSE118413), ATAC-Seq (GSE104720), and DNase1-Seq (GSE61793). FAIRE-Seq peaks were downloaded from the UCSC genome browser and not re-called.

### Chromatin interaction data

To assess chromatin looping, we utilized a previously published H3K27ac HiChIP dataset from a normal hTERT-immortalized endometrial cell line (E6E7hTERT) and three endometrial cancer cell lines (ARK1, Ishikawa and JHUEM-14) (*O'Mara et al., 2019*). This study identified 66,092 to 449,157 *cis* HiChIP loops (5 kb–2 Mb in length) per cell line. The majority of loops involved interactions of over 20 kb in distance, 35–40% of loops had contact with a promoter and those promoter-associated loops had a median span >200 kb. Contact data were from the original publication and not re-called for this study. Note that *Figures 2C* and *4C* were made using a combination of the UCSC genome browser to map the location of regions of open chromatin and ChIP-Seq peaks and Illustrator to simplify the images from the genome browser.

## Cell culture and *HAND2* knockdown

Human hTERT-immortalized endometrial stromal fibroblasts (T-HESC; CRL-4003, ATCC) were grown in maintenance medium, consisting of Phenol Red-free DMEM (31053–028, Thermo Fisher Scientific),

supplemented with 10% charcoal-stripped fetal bovine serum (CS-FBS; 12676029, Thermo Fisher Scientific), 1% L-glutamine (25030–081, Thermo Fisher Scientific) , 1% sodium pyruvate (11360070, Thermo Fisher Scientific), and 1x insulin-transferrin-selenium (ITS; 41400045, Thermo Fisher Scientific). A total of $2 \times 10^5$ cells were plated per well of a six-well plate and 18 hr later cells in 1750 µl of Opti-MEM (31985070, Thermo Fisher Scientific) were transfected with 50 nM of siRNA targeting *HAND2* (s18133; Silencer Select Pre-Designed siRNA; cat # 4392420, Thermo Fisher Scientific) and 9 µl of Lipofectamine RNAiMAX (133778–150, Invitrogen) in 250 µl Opti-MEM. BlockIT Fluorescent Oligo (44–2926, Thermo Fisher Scientific) was used as a scrambled non-targeting RNA control. Cells were incubated in the transfection mixture for 6 hr. Then, cells were washed with PBS and incubated in the maintenance medium overnight. Cells in the control wells were checked under the microscope for fluorescence the next day. Forty-eight hr post-treatment, cells were washed with PBS, trypsinized (0.05% Trypsin-EDTA; 15400–054, Thermo Fisher Scientific) and total RNA was extracted using RNeasy Plus Mini Kit (74134, QIAGEN) following the manufacturer's protocol. The knockdown experiment was done in three biological replicates. To test for the efficiency of the knockdown, cDNA was synthesized from 100 to 200 ng RNA using Maxima H Minus First Strand cDNA Synthesis Kit (K1652, Thermo Fisher Scientific) following the manufacturer's protocol. qRT-PCR was performed using QuantiTect SYBR Green PCR (204143, QIAGEN). *HAND2* primers: forward CACCAGCTACATCGCC TACC, reverse ATTTCGTTCAGCTCCTTCTTCC. *GAPDH* housekeeping gene was used for normalization; primers forward AATCCCATCACCATCTTCCA, reverse TGGACTCCACGACGTACTCA. Samples that showed >70% knockdown efficiency were used for RNA-Seq.

## *HAND2* knockdown RNA-Seq analysis

RNA from knockdown and control samples were DNase treated with TURBO DNA-free Kit (AM1907, Thermo Fisher Scientific) and RNA quality and quantity were assessed on 2100 Bioanalyzer (Agilent Technologies, Inc). RNA-Seq libraries were prepared using TruSeq Stranded Total RNA Library Prep Kit with Ribo-Zero Human (RS-122–2201, Illumina Inc) following manufacturer's protocol. Library quality and quantity were checked on 2100 Bioanalyzer and the pool of libraries was sequenced on Illumina HiSEQ4000 (single-end 50 bp) using manufacturer's reagents and protocols. Quality control, Ribo-Zero library preparation and Illumina sequencing were performed at the Genomics Facility at The University of Chicago.

All sequencing data were uploaded and analyzed on the Galaxy platform (https://usegalaxy.org/; Version 20.01). Individual reads for particular samples were concatenated using the 'Concatenate datasets' tool (version 1.0.0). We used HISAT2 (version 2.1.0+galaxy5) (*Kim et al., 2015*) to align reads to the Human hg38 reference genome using 'Single-end' option, and reporting alignments tailored for transcript assemblers including StringTie. Transcripts were assembled and quantified using StringTie (v1.3.6) (*Pertea et al., 2015*; *Pertea et al., 2016*), with reference file to guide assembly and the 'reference transcripts only' option, and output count files for differential expression with DESeq2/edgeR/limma-voom. Differentially expressed genes were identified using DESeq2 (version 2.11.40.6+galaxy1; *Anders and Huber, 2010*; *Love et al., 2014*). The reference file for StringTie guided assembly was wgEncodeGencodeBasicV33. The volcano plot was generated using *Blighe et al., 2020*.

## Trans-well migration assay

Human hTERT-immortalized ESFs (CRL-4003, ATCC) were selected as a model ESF cell line because they are proliferative, maintain hormone responsiveness, and gene expression patterns characteristic of primary ESFs, and have been relatively well characterized (*Krikun et al., 2004*). ESFs were cultured in the maintenance medium as described above in T75 flasks until ~80% confluent. Cryopreserved primary adult human CD56+ NK cells purified by immunomagnetic bead separation were obtained from ATCC (PCS-800–019) and cultured in RPMI-1640 containing 10% FBS and 500 IU/ml IL2 in T75 flasks for 2 days prior to trans-well migration assays. We used the immortalized first trimester extravillous trophoblast cell line HTR-8/SVneo (*Graham et al., 1993*), because it maintains characteristics of extravillous trophoblasts and has previously been shown to be a good model of trophoblast migration and invasion (*Iacob et al., 2008*; *Paiva et al., 2009*; *Hannan et al., 2010*). HTR-8/SVneo cells were obtained from ATCC (CRL-3271) and cultured in RPMI-1640 containing 5%

FBS in T75 flasks for 2 days prior to trans-well migration assays. All cell lines were determined to be mycoplasma free before each experiment.

A total of $3 \times 10^4$ ESFs were plated per well of a 24-well plate and 18 hr later cells were transfected in Opti-MEM with 10 nM (per well) of siRNA targeting *HAND2* (s18133; Silencer Select Pre-Designed siRNA, cat # 4392420; Thermo Fisher Scientific) or *IL15* (s7377; Silencer Select Pre-Designed siRNA, cat # 4392420; Thermo Fisher Scientific) and 1.5 µl (per well) of Lipofectamine RNAiMAX (133778–150; Invitrogen). As a negative control we used Silencer Select negative control No. 1 (4390843; Thermo Fisher Scientific). ESFs were incubated in the transfection mixture for 6 hr. Then, ESFs were washed with warm PBS and incubated in the maintenance medium overnight. Efficiency of the knockdown was confirmed 48 hr post-treatment by qRT-PCR, media from each well was transferred to new 24-well plates and stored at 4℃. Total RNA from cells was extracted using RNeasy Plus Mini Kit (74134, QIAGEN) following the manufacturer's protocol. cDNA was synthesized from 10 ng RNA using Maxima H Minus First Strand cDNA Synthesis Kit (K1652, Thermo Fisher Scientific) following the manufacturer's protocol. qRT-PCR was performed using TaqMan Fast Universal PCR Master Mix 2X (4352042, Thermo Fisher Scientific), with primers for *HAND2* (Hs00232769_m1), *IL15* (Hs01003716_m1), and *Malat* (Hs00273907_s1) as a control housekeeping gene. Conditioned media from samples with >70% knockdown efficiency was used for trans-well migration assays.

Corning HTS Trans-well permeable supports were used for the trans-well migration assay (Corning, cat # CLS3398). Prior to the assays, 500 µl of media conditioned for 12 hr was centrifuged for 3 min at 1000 RPM to pellet any cells. For experiments using recombinant human IL15 (rhIL15), we added 10 ng/ml (AbCam, ab259403) of rhIL15 to fresh, non-conditioned ESF media; 10 ng/ml has previously been shown to induce migration of JEG-3 choriocarcinoma cells (*Zygmunt et al., 1998*). For neutralizing antibody experiments, either 1 µg/ml of anti-IL15 IgG (AbCam cat # MA5-23729) or control IgG (AbCam cat # 31903) were added to non-conditioned ESF media; 1 µg/ml has previously been shown to neutralize ESF-derived proteins and inhibit AC-1M88 trophoblast cell migration in trans-well assays (*Gellersen et al., 2010*; *Gellersen et al., 2013*). Plates were incubated with shaking at 37℃ for 30 min prior to initiation of migration assays. During antibody incubation, NK and HTR-8/SVneo cells were collected and resuspended in fresh ESF growth media. For the trans-well migration assay, $5 \times 10^6$ of either NK or HTR-8/SVneo cells were added to each well of the upper chamber and either treatment or control media were added to the lower chambers. Plates were incubated at 37℃ with 5% $CO_2$.

After 8 hr incubation, we removed the upper plate (containing remaining NK and HTR-8/SVneo cells) and discarded non-migrated cells. Fifty µl from each well in the lower chamber was transferred into a single well of a 96-well opaque plate. We used the CellTiter-Glo luminescent cell viability assay (G7570, Promega) to measure luminescence, which is proportional to the number of live cells per well. Data are reported as effect sizes (mean differences) between treatment and control. Confidence intervals are bias-corrected and accelerated. The p-values reported are the likelihoods of the observed effect sizes, if the null hypothesis of zero difference is true and calculated from a two-sided permutation t-test (5000 reshuffles of the control and test labels). Cumming estimation plots and estimation statistics were calculated using DABEST R package (*Ho et al., 2019*).

## Acknowledgements

The authors thank the following researchers for providing pregnant endometrial samples: GP Wagner (Yale University) – *Monodelphis domestica*; RR Behringer (The University of Texas MD Anderson Cancer Center) – *Carollia perspicillata*; BC Paria (Vanderbilt University School of Medicine) – *Mesocricetus auratus*, *Mus musculus*; AT Fazleabas (Michigan State University) – *Papio anubis*; DK Merriman (University of Wisconsin Oshkosh) – *Ictidomys tridecemlineatus*. We are also grateful to AM Bamberger (University Hospital Eppendorf) for providing the trans-well migration assay protocol, D Glubb (QIMR Berghofer Medical Research Institute) for assistance in interpreting the HiChIP assay data, R Beaumont (University of Exeter Medical School) and RM Freathy (University of Exeter) for assistance with interpreting maternal-fetal birth weight GWAS data, and VL Hansen (University of New Mexico) for providing details on stages of opossum RNA-Seq data. VJL thanks the Department of Human Genetics at The University of Chicago for support during the planning and preliminary data generation phase of this work. MM thanks Michael Sulak for the help with editing the manuscript.

## Additional information

### Funding

| Funder | Grant reference number | Author |
|---|---|---|
| Burroughs Wellcome Fund | Preterm Birth Initiative 1013760 | Vincent J Lynch |
| March of Dimes Foundation | UChicago-Northwestern-Duke Prematurity Research Center | Vincent J Lynch |

The funders had no role in study design, data collection and interpretation, or the decision to submit the work for publication.

### Author contributions

Mirna Marinić, Conceptualization, Data curation, Formal analysis, Investigation, Writing - original draft, Writing - review and editing; Katelyn Mika, Conceptualization, Data curation, Formal analysis, Methodology, Writing - original draft, Writing - review and editing; Sravanthi Chigurupati, Formal analysis, Writing - original draft; Vincent J Lynch, Conceptualization, Data curation, Formal analysis, Supervision, Funding acquisition, Investigation, Visualization, Writing - original draft, Project administration, Writing - review and editing

### Author ORCIDs

Mirna Marinić (iD) https://orcid.org/0000-0002-7037-8389
Katelyn Mika (iD) https://orcid.org/0000-0002-2170-9364
Vincent J Lynch (iD) https://orcid.org/0000-0001-5311-3824

### Decision letter and Author response

Decision letter https://doi.org/10.7554/eLife.61257.sa1
Author response https://doi.org/10.7554/eLife.61257.sa2

## Additional files

### Supplementary files

• Transparent reporting form

### Data availability

Sequencing data have been deposited in GEO under accession codes GSE155170 and GSE155322.

The following datasets were generated:

| Author(s) | Year | Dataset title | Dataset URL | Database and Identifier |
|---|---|---|---|---|
| Mika K, Lynch VJ | 2021 | Mouse Endometrium Individual 1 | https://www.ncbi.nlm.nih.gov/geo/query/acc.cgi?acc=GSM5100045 | NCBI Gene Expression Omnibus, GSM5100045 |
| Mika K, Lynch VJ | 2021 | Mouse Endometrium Individual 2 | https://www.ncbi.nlm.nih.gov/geo/query/acc.cgi?acc=GSM5100046 | NCBI Gene Expression Omnibus, GSM5100046 |
| Mika K, Lynch VJ | 2021 | Mouse Endometrium Individual 3 | https://www.ncbi.nlm.nih.gov/geo/query/acc.cgi?acc=GSM5100047 | NCBI Gene Expression Omnibus , GSM5100047 |
| Mika K,  Lynch VJ | 2020 | Baboon Endometrium Individual 1 | https://www.ncbi.nlm.nih.gov/geo/query/acc.cgi?acc=GSM4696515 | NCBI Gene Expression Omnibus, GSM4696515 |
| Mika K,  Lynch VJ | 2020 | Baboon Endometrium Individual 2 | https://www.ncbi.nlm.nih.gov/geo/query/acc.cgi?acc=GGSM4696516 | NCBI Gene Expression Omnibus, GGSM4696516 |

| | | | | | |
|---|---|---|---|---|---|
| Mika K, | Lynch VJ | 2020 | Baboon Endometrium Individual 3 | https://www.ncbi.nlm.nih.gov/geo/query/acc.cgi?acc=GSM4696517 | NCBI Gene Expression Omnibus, GSM4696517 |
| Mika K, | Lynch VJ | 2020 | Hamster Endometrium Individual 1 Replicate 1 | https://www.ncbi.nlm.nih.gov/geo/query/acc.cgi?acc=GSM4696518 | NCBI Gene Expression Omnibus, GSM4696518 |
| Mika K, | Lynch VJ | 2020 | Hamster Endometrium Individual 2 Replicate 1 | https://www.ncbi.nlm.nih.gov/geo/query/acc.cgi?acc=GSM4696519 | NCBI Gene Expression Omnibus, GSM4696519 |
| Mika K, | Lynch VJ | 2020 | Hamster Endometrium Individual 3 Replicate 1 | https://www.ncbi.nlm.nih.gov/geo/query/acc.cgi?acc=GSM4696520 | NCBI Gene Expression Omnibus, GSM4696520 |
| Mika K, | Lynch VJ | 2020 | Hamster Endometrium Individual 1 Replicate 2 | https://www.ncbi.nlm.nih.gov/geo/query/acc.cgi?acc=GSM4696521 | NCBI Gene Expression Omnibus, GSM4696521 |
| Mika K, | Lynch VJ | 2020 | Hamster Endometrium Individual 2 Replicate 2 | https://www.ncbi.nlm.nih.gov/geo/query/acc.cgi?acc=GSM4696522 | NCBI Gene Expression Omnibus, GSM4696522 |
| Mika K, | Lynch VJ | 2020 | Hamster Endometrium Individual 3 Replicate 2 | https://www.ncbi.nlm.nih.gov/geo/query/acc.cgi?acc=GSM4696523 | NCBI Gene Expression Omnibus, GSM4696523 |
| Mika K, | Lynch VJ | 2020 | Bat Endometrium Individual 1 | https://www.ncbi.nlm.nih.gov/geo/query/acc.cgi?acc=GSM4696524 | NCBI Gene Expression Omnibus, GSM4696524 |
| Mika K, | Lynch VJ | 2020 | Bat Endometrium Individual 2 | https://www.ncbi.nlm.nih.gov/geo/query/acc.cgi?acc=GSM4696525 | NCBI Gene Expression Omnibus, GSM4696525 |
| Mika K, | Lynch VJ | 2020 | Squirrel Endometrium Individual 1 | https://www.ncbi.nlm.nih.gov/geo/query/acc.cgi?acc=GSM4696526 | NCBI Gene Expression Omnibus, GSM4696526 |
| Mika K, | Lynch VJ | 2020 | Squirrel Endometrium Individual 2 | https://www.ncbi.nlm.nih.gov/geo/query/acc.cgi?acc=GSM4696527 | NCBI Gene Expression Omnibus, GSM4696527 |
| Marinić M, Lynch VJ | | 2020 | siRNA Ctrl_rep1a | https://www.ncbi.nlm.nih.gov/geo/query/acc.cgi?acc=GSM4699113 | NCBI Gene Expression Omnibus, GSM4699113 |
| Marinić M, Lynch VJ | | 2020 | siRNA Ctrl_rep1b | https://www.ncbi.nlm.nih.gov/geo/query/acc.cgi?acc=GSM4699114 | NCBI Gene Expression Omnibus, GSM4699114 |
| Marinić M, Lynch VJ | | 2020 | siRNA Ctrl_rep2a | https://www.ncbi.nlm.nih.gov/geo/query/acc.cgi?acc=GSM4699115 | NCBI Gene Expression Omnibus, GSM4699115 |
| Marinić M, Lynch VJ | | 2020 | siRNA Ctrl_rep2b | https://www.ncbi.nlm.nih.gov/geo/query/acc.cgi?acc=GSM4699116 | NCBI Gene Expression Omnibus, GSM4699116 |
| Marinić M, Lynch VJ | | 2020 | siRNA Ctrl_rep3a | https://www.ncbi.nlm.nih.gov/geo/query/acc.cgi?acc=GSM4699117 | NCBI Gene Expression Omnibus, GSM4699117 |
| Marinić M, Lynch VJ | | 2020 | siRNA Ctrl_rep3b | https://www.ncbi.nlm.nih.gov/geo/query/acc.cgi?acc=GSM4699118 | NCBI Gene Expression Omnibus, GSM4699118 |
| Marinić M, Lynch VJ | | 2020 | siRNA HAND2_rep1a | https://www.ncbi.nlm.nih.gov/geo/query/acc.cgi?acc=GSM4699119 | NCBI Gene Expression Omnibus, GSM4699119 |
| Marinić M, Lynch VJ | | 2020 | siRNA HAND2_rep1b | https://www.ncbi.nlm.nih.gov/geo/query/acc.cgi?acc=GSM4699120 | NCBI Gene Expression Omnibus, GSM4699120 |
| Marinić M, Lynch VJ | | 2020 | siRNA HAND2_rep2a | https://www.ncbi.nlm.nih.gov/geo/query/acc.cgi?acc=GSM4699121 | NCBI Gene Expression Omnibus, GSM4699121 |
| Marinić M, Lynch VJ | | 2020 | siRNA HAND2_rep2b | https://www.ncbi.nlm.nih.gov/geo/query/acc. | NCBI Gene Expression Omnibus, GSM4699122 |

| | | | cgi?acc=GSM4699122 | |
|---|---|---|---|---|
| Marinić M, Lynch VJ | 2020 | siRNA HAND2_rep3a | https://www.ncbi.nlm. nih.gov/geo/query/acc. cgi?acc=GSM4699123 | NCBI Gene Expression Omnibus, GSM4699123 |
| Marinić M, Lynch VJ | 2020 | siRNA HAND2_rep3b | https://www.ncbi.nlm. nih.gov/geo/query/acc. cgi?acc=GSM4699124 | NCBI Gene Expression Omnibus, GSM4699124 |
| Mika K, Lynch VJ | 2020 | Evolutionary transcriptomics implicates HAND2 in the origins of implantation and regulation of gestation length | https://www.ncbi.nlm. nih.gov/geo/query/acc. cgi?acc=GSE155170 | NCBI Gene Expression Omnibus, GSE155170 |
| Marinić M, Lynch VJ | 2020 | Evolutionary transcriptomics implicates HAND2 in the origins of implantation and regulation of gestation length | https://www.ncbi.nlm. nih.gov/geo/query/acc. cgi?acc=GSE155322 | NCBI Gene Expression Omnibus, GSE155322 |

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
