## [Decision Letter]

**Acceptance summary:**

Parsing mechanisms of disease from the perspective of evolutionary biology is a powerful approach. The manuscript by Marinić et al. uses an innovative gene expression dataset in an evolutionary framework to identify a set of transcripts whose endometrial expression emerged at the eutherian mammal stem lineage. One of these transcripts is for the transcription factor HAND2. Using both existing datasets and experimental data the authors build a model of the activity of HAND2 and its associated protein IL15 at the maternal-fetal interface and implicate the proteins in both the evolution and disorders of pregnancy. The work illustrates the utility of evolutionary analysis for elucidating functional mechanisms of complex disorders and substantially contributes to our knowledge of the evolution and diseases of pregnancy.

**Decision letter after peer review:**

Thank you for submitting your article "Evolutionary transcriptomics implicates HAND2 in the origins of implantation and regulation of gestation length" for consideration by *eLife*. Your article has been reviewed by three peer reviewers, and the evaluation has been overseen by a Reviewing Editor and George Perry as the Senior Editor. The following individual involved in review of your submission has agreed to reveal their identity: Abigail LaBella (Reviewer #2).

The reviewers have discussed the reviews with one another and the Reviewing Editor has drafted this decision to help you prepare a revised submission.

Summary:

Parsing mechanisms of disease from the perspective of evolutionary biology is a powerful approach. The manuscript by Marinić et al. uses an innovative expression dataset in an evolutionary framework to identify a set of transcripts whose endometrial expression emerged at the eutherian stem lineage. One of these is the transcription factor HAND2. Using both existing datasets and experimental data the authors build a model of the activity of HAND2 and its associated protein IL15 at the maternal-fetal interface and implicate the proteins in both the evolution and disorders of pregnancy. The work illustrates the utility of evolutionary analysis for elucidating functional mechanisms of complex disorders and substantially contributes to our knowledge of the evolution and diseases of pregnancy.

Revisions:

1) Figure 1C appears interesting but there is no comparison or controls. Without comparison, for example the histotrophic phase, it appears difficult to conclude that estrogen signaling genuinely persists during pregnancy in the opossum. pESR1 staining in the tissue section is ubiquitous with no evidence of nuclear localisation, raising concerns about antibody specificity. KI67 staining may be more informative?

2) The authors used a large single-cell RNA-seq data set to map HAND2 expression at the human maternal-fetal interface in the first-trimester of pregnancy (Vento-Tormo et al., 2018). They demonstrate that HAND2 expression is confined to 3 maternal subsets, termed endometrial stromal fibroblast (ESF) 1 and 2 and decidual stromal cells (DSC). If we are not mistaken, in the Vento-Tormo paper, these populations of cells were labelled decidual stromal cells 1-3 (DS1-3), emphasizing that all these cells were decidualized, as expected in pregnancy. Vento-Tormo et al. further demonstrated that the differences in gene expression between DS subsets relate to their topography in the maternal tissue. Hence, it is confusing that the authors changed the terminology of these subsets, giving the erroneous impression of two undifferentiated ESF population and a single DS/DSC population in pregnancy. By doing so, the inference seems to be that T-HESC, a telomerase-transformed endometrial stromal cell line used in functional studies, is a good model of ESF populations *in vivo*, which is doubtful.

3) Figure 2G. The authors state that “We also used previously published gene expression datasets (see Materials and methods) to explore if HAND2 was associated with disorders of pregnancy and found significant HAND2 dysregulation in the endometria of women with infertility (IF) and recurrent spontaneous abortion (RSA) compared to fertile controls” – This bold statement is based on microanalysis of merely 5 biopsies in each group. Considering the intrinsic temporo-spatial heterogeneity of the cycling endometrium, this sample size is grossly inadequate. The microarray study was published in 2011. In fact there are several more recent and more robust datasets available (e.g. 115 IF biopsies in GSE58144 and 20 RM biopsies in GPL11154) that should instead be used. These comments also apply to Figure 4G.

4) The authors also state “HAND2 was not differentially expressed in ESFs or DSCs from women with preeclampsia (PE) compared to controls (Figure 2G).” It is unclear which dataset this was based on. The authors' claim seem to indicate that this was single-cell data? In any case, the sample size is again grossly inadequate to draw robust conclusions without further validation in a much larger cohort of samples.

5) Figure 3. The authors decided to knockdown HAND2 in T-HESC, a telomerase-transformed endometrial stromal cell line, and performed RNA-seq 48 h later. The cells were not decidualized or even treated with progesterone. Hence, the rationale for this experiment, and its relevance to the *in vivo* situation, is genuinely lost on me. See also comment regarding the renaming of DS subsets into ESF. In an undifferentiated state, these cells are not representative of gestational cells (with the possible exception that decidual senescence is characterized by progesterone resistance, i.e. re-activation of genes that are suppressed by progesterone). More importantly, as HAND2 is critical for the identity of these cells, perhaps knockdown triggers a stress response? For example, from the data presented in Figure 3—source data 1 (it would be helpful to add gene names), on of the strongest up-regulated gene upon HAND2 knockdown is BLCAP2 [Log2(FC): 10.2], which encodes a protein that reduces cell growth by stimulating apoptosis.

6) The authors illustrated the importance of examining the right cellular state: knockdown HAND2 in T-HESC increases IL15 expression whereas it is well established that HAND2 knockdown in decidual cells decreases IL15 expression. Further, IL15 is strongly induced upon decidualization and previous studies on primary endometrial stromal cells demonstrated that IL15 secretion is undetectable in undifferentiated cells whereas it is abundantly secreted upon decidualization (PMID: 31965050). Thus, we suggest that the should repeat HAND2 KD in decidualizing T-HESC and measure IL15 secretion in both states, with and without HAND2 knockdown, in future experiments.

7) Figure 3B – it is unclear what is compared here: genes deregulated upon HAND2 knockdown in T-HESC versus knockdown NR2F2, FOXO1 and GAT2 in decidualized primary cultures? If this is the case, the comparison is not informative as it involves two different cell states. It is surprising that FOSL2 was not included in this analysis.

8) We do not understand the relevance of the experiments described in Figure 5 to the context of gestation length or preterm birth. Trophoblast invasion will have been completed in the second trimester of pregnancy – what is the purpose/message of these experiments? What is the level of IL15 secreted by these cells? Again the T-HESC appear not decidualized – so, what is the relevance to either the midluteal implantation window or gestation?

9) What is the evolution of IL15 expression at the maternal-fetal interface? Does it parallel HAND2?

10) Of the 149 genes that unambiguously evolved endometrial expression why was only HAND2 analyzed? We are not suggesting that each gene be followed up with this level of rigor but would you hypothesize that each of the genes you identified play a role in eutherian reproduction? Or are there other major innovations that some of these genes may be associated with? How frequently would this pattern occur by chance?

11) Figures 2F and 4F – there appears to be a gap in the data points during the third trimester (which looks like it says "thirdr"). Is there still a negative trend if each section is analyzed independently as if they were independent datasets? Aka could this linear trend be composed of two separate trends instead?

12) Please provide the binary encoded data used for this analysis as it could be readily used by other research groups for similar analysis. The custom database of genes implicated in preterm birth would also be a useful dataset.

13) It was helpful to hear from one of the authors that the known HAND2 gene wasn't knocked out in mice, so it was an easy early pregnancy gene to start with. Perhaps this should be stated in the revised manuscript?

14) To reproduce the study, there were a couple of questions around the production of the conditioned media including, how long were the cells incubated in the media and what was the volume of the media use. Please include this information in the revised manuscript.

15) Can you further explain why the opossum was used to measure the estrogen levels?

16) The relationship between ESR1 and HAND2 is a little unclear. Is ESR1 expression correlated with HAND2 expression in all species studied?

---

## [Author Response]

Revisions:1) Figure 1C appears interesting but there is no comparison or controls. Without comparison, for example the histotrophic phase, it appears difficult to conclude that estrogen signaling genuinely persists during pregnancy in the opossum.

We agree that showing expression of *HAND2*, *ESR1*, and estrogen responsive genes at a single time point can make it difficult to conclude that estrogen signaling persists throughout opossum pregnancy, particularly in the absence of a “control”. Therefore, we have updated Figure 1C (now Figure 1D) to include RNA-Seq data from the uterus of non-pregnant control, pregnancy day 7, pregnancy day 12.5, pregnancy day 13, as well as pregnancy days 13.5-14.5 (late placental to early parturition phase) and 9-10 months post-partum. Note that previously we only showed data for day 12.5 (just after hatching), and that birth occurs on day 14.5. These new data are consistent with our claim that RNA-Seq data indicate persistent estrogen signaling during opossum pregnancy, for example, showing that *ESR1*, *FGF9*, and *WNTs* are expressed across all stages. A previous study has shown that *ESR1* is an estrogen responsive gene in wallaby (Renfree and Blanden, 2000), providing an additional evidence for estrogen signaling during pregnancy in Marsupials. We have updated the manuscript to include a description of the new time course data, as well as indicate that *ESR1* is an estrogen responsive gene in Marsupials (Renfree and Blanden, 2000).

More interestingly, these data also show that *HAND2* expression is dynamic throughout pregnancy, and is highest in the non-pregnant uterus, down-regulated during mid- and late-pregnancy (below our expression cutoff of TPM=2), and increases in expression at 14d about 12 hours before parturition. We interpret these data to support our conclusions that *HAND2* is generally not expressed during either the histotrophic or placental phase, and reinforces our conclusion that estrogen signaling (as evidenced by the expression of estrogen responsive genes such as *ESR1*) persists during pregnancy in the opossum. The coincidence of *ESR1* and *HAND2* down-regulation suggests that the negative regulatory interaction between *HAND2* and *ESR1* does not occur in opossum (if it did, as *HAND2* expression decreased, *ESR1* expression would increase). We have updated the manuscript to reflect this new finding. We thank the reviewers for these very useful suggestions.

pESR1 staining in the tissue section is ubiquitous with no evidence of nuclear localization, raising concerns about antibody specificity. KI67 staining may be more informative?

Previous studies have shown that progesterone and estrogen receptor proteins are expressed in the endometrium throughout pregnancy in Marsupials, and down-regulated in mid- and late-pregnancy (which is also reflected in our RNA-Seq time course shown in previous Figure 1C, now Figure 1D). Thus, our report of estrogen receptor protein expression in the pregnant opossum endometrium is supported by previous studies. We now include reference for those studies, demonstrating estrogen receptor expression in the pregnant endometrium of the brush-tailed possum (*Trichosurus vulpecula*) (Young and McDonald, 1982; Curlewis and Stone, 1982) and tammar wallaby (*Macropus eugenii*) (Renfree and Blanden, 2000). While we believe the staining is specific, we appreciate the concern that the pESR1 staining may not be specific and have therefore moved the immunohistochemistry figure from the main text to a figure supplement. We do not believe this alters the conclusions from this figure because the RNA-Seq time course data indicate that estrogen-responsive genes are expressed throughout pregnancy in opossum. Unfortunately, we were unable to find an antibody that cross-reacted with *Monodelphis* KI67, however, it is expressed just above our TPM=2 cutoff in both *Monodelphis* non-pregnant and pregnant endometria. These data are now shown in Figure 1D.

2) The authors used a large single-cell RNA-seq data set to map HAND2 expression at the human maternal-fetal interface in the first-trimester of pregnancy (Vento-Tormo et al., 2018). They demonstrate that HAND2 expression is confined to 3 maternal subsets, termed endometrial stromal fibroblast (ESF) 1 and 2 and decidual stromal cells (DSC). If we are not mistaken, in the Vento-Tormo paper, these populations of cells were labelled decidual stromal cells 1-3 (DS1-3), emphasizing that all these cells were decidualized, as expected in pregnancy. Vento-Tormo et al. further demonstrated that the differences in gene expression between DS subsets relate to their topography in the maternal tissue. Hence, it is confusing that the authors changed the terminology of these subsets, giving the erroneous impression of two undifferentiated ESF population and a single DS/DSC population in pregnancy. By doing so, the inference seems to be that T-HESC, a telomerase-transformed endometrial stromal cell line used in functional studies, is a good model of ESF populations *in vivo*, which is doubtful.

While we appreciate the concerns of the reviewers, the naming used by Vento-Tormo et al. (2018) based on the location of these three cell populations in the decidua does not reflect their cell-type identity. The authors identified five populations of cells in the endometrial stromal lineage, including two perivascular populations (likely reflecting the mesenchymal stem cell-like progenitor of endometrial stromal fibroblasts and decidual stromal cells) and three cell types they call “decidual stromal cells” and label “dS1-3”. However, based on the gene expression patterns of “dS1-3” (shown in Vento-Tormo *et al.* Figure 3a), only “dS3” are decidualized, as indicated by expression of classical markers of decidualization such and *PRL* (Tabanelli, Tang and Gurpide, 1992) and *IGFBP1/2/6* (Tabanelli, Tang and Gurpide, 1992; Kim, Jaffe and Fazleabas, 1999). In stark contrast, “dS1” do not express decidualization markers but highly express markers of endometrial stromal fibroblasts (ESFs) such as *TAGLN* and *ID2,* as well as markers of proliferating ESFs including *ACTA2* (Kim, Jaffe and Fazleabas, 1999). “dS2” also express ESFs markers (*TAGLN*, *ID2*, *ACTA2*), but additionally *LEFTY2* and *IGFBP1/2/6*, consistent with ESFs that have initiated the process of decidualization. These data indicate that the “dS1” and “dS2” populations are both ESFs, but “dS2” are ESFs that have initiated decidualization (because they express *IGFBPs* but not *PRL*), and that “dS3” are DSCs. Vento-Tormo *et al.* show that the differences in gene expression between “dS1-3” are related to their topography in the endometrium, but degree of decidualization (“dS1”/ESF1 <“dS2”/ESF2 < “dS3”/DSC) is *also* linked to differential gene expression.

Consistent with this, other scRNA-Seq studies have identified two ESF populations and one DSC population in the first trimester decidua, and used pseudotime analyses to show that they represent different states of differentiation from ESFs to mature DSCs (Suryawanshi et al., 2018). Therefore, we prefer to use the ESF1/ESF2/DSC nomenclature because it more accurately reflects the biology and gene expression profile of these cell-types than the “dS1-3” naming convention.

Importantly, ESFs are present in the endometrium during the first trimester and persist in the endometrium until term (Sakabe et al., 2020; Richards et al., 1995; Munoz-Fernandez et al., 2019; Suryawanshi et al., 2018) – we apologize if this was unclear in the manuscript. Our naming of the “dS1” and “dS2” populations as ESF1 and ESF2 makes it clear that there are ESFs in the decidua during pregnancy and that the T-HESC cell line is a good model of these *in vivo* ESF populations. We appreciate that changing the terminology for these cell types can be confusing, and have added an explanation for our name change in the Materials and methods section describing the Vento-Tormo *et al.* We also indicate in the Results section that readers “see Materials and methods for cell-type naming convention” and to explain that ESFs persist in the endometrium till term. We hope this helps allay the concerns of the reviewers.

3) Figure 2G. The authors state that “We also used previously published gene expression datasets (see Materials and methods) to explore if HAND2 was associated with disorders of pregnancy and found significant HAND2 dysregulation in the endometria of women with infertility (IF) and recurrent spontaneous abortion (RSA) compared to fertile controls” – This bold statement is based on microanalysis of merely 5 biopsies in each group. Considering the intrinsic temporo-spatial heterogeneity of the cycling endometrium, this sample size is grossly inadequate. The microarray study was published in 2011. In fact there are several more recent and more robust datasets available (e.g. 115 IF biopsies in GSE58144 and 20 RM biopsies in GPL11154) that should instead be used. These comments also apply to Figure 4G.

While we agree that there can be significant temporo-spatial heterogeneity across endometrial samples, the sampling protocol for GSE58144 makes it more difficult to analyze than the GSE26787 dataset that we used, even though GSE26787 has fewer samples per condition (five endometrial biopsies in non-conceptional cycles in the mid-luteal phase). For example, RNA for the GSE58144 dataset was collected from mid-luteal phase endometrial biopsies between 2006 and 2013. Thus, while the GSE58144 data were published more recently than the GSE26787 data, at least some GSE58144 samples were collected earlier and it is not clear from the available information how the samples were stored or batch-processed on the Agilent microarray used to quantify gene expression. Unfortunately, this leads to difficulties in data analysis – for example, it is not clear how to appropriately batch-correct these samples. Based on the GEO submission information, there are at least two “cohorts”, but no genes are significantly differentially expressed (DE) among either cohort 1 or cohort 2 at FDR≤0.10, and only two genes are significantly DE at FDR≤0.10 (*AQP11* and *KYNU*) when cohorts 1 and 2 are combined. Thus, while GSE58144 is much larger than the dataset we utilized, their study design reduces power to detect significantly DE genes. To reflect the reviewer’s concerns, we have added a caveat to the section of these results, which states that: “Although sample sizes of these datasets are small, and the intrinsic temporo-spatial heterogeneity of the endometrium remains a potential confounding factor, we found that *HAND2* was dysregulated in the endometria of women with implantation failure (IF) and recurrent spontaneous abortion (RSA), while it was not differentially expressed in ESFs or DSCs from women with preeclampsia (PE), compared to controls (Figure 2G).”

We were unable to find a dataset of 20 recurrent miscarriage biopsies because the accession GPL11154 refers to all datasets generated on the Illumina HiSeq 2000 (Homo sapiens) platform and the search for “GEO (recurrent miscarriage) AND "*Homo sapiens*"[porgn: txid9606]” did not identify an RM dataset with 20 biopsies.

4) The authors also state “HAND2 was not differentially expressed in ESFs or DSCs from women with preeclampsia (PE) compared to controls (Figure 2G).” It is unclear which dataset this was based on. The authors' claim seem to indicate that this was single-cell data? In any case, the sample size is again grossly inadequate to draw robust conclusions without further validation in a much larger cohort of samples.

The expression of *HAND2* and *IL5* in ESFs and DSCs from women with normal pregnancies and severe preeclampsia was assessed from a previously generated Agilent Whole Human Genome Microarray 4x44K v2 dataset (GSE91077), not from scRNA-Seq data. We have included this information in the Materials and methods section to ensure it is clear which technology was used to generate this dataset. Also see the answer to the previous question.

5) Figure 3. The authors decided to knockdown HAND2 in T-HESC, a telomerase-transformed endometrial stromal cell line, and performed RNA-seq 48 h later. The cells were not decidualized or even treated with progesterone. Hence, the rationale for this experiment, and its relevance to the *in vivo* situation, is genuinely lost on me.

Previous studies have already explored the functions of *HAND2* and *IL15* in DSCs, including a recent study showing that *IL15* is a direct target gene of HAND2 in DSCs (Murata et al., 2020). Both *HAND2* and *IL15* are expressed in ESFs, including ESFs at the maternal-fetal inference in the Vento-Tormo et al. (2018) dataset. Thus, the aim of our experiments was to determine if *HAND2* also regulates *IL15* in ESFs, and if so, what the consequences of *IL15* expression in ESFs might be (Figure 5). We apologize if this rationale was not clear in the manuscript, and recognize that the differences in the “dS1-3” vs. “ESF1-2/DSC” nomenclatures may have contributed to the confusion. We have edited the text to make it explicit that “dS1-2” are “ESF1-2”, that ESFs are present at the maternal-fetal interface until term, and therefore the rationale for our experiment was to explore the functions of *HAND2* and *IL15* in ESFs at the maternal-fetal interface.

See also comment regarding the renaming of DS subsets into ESF. In an undifferentiated state, these cells are not representative of gestational cells (with the possible exception that decidual senescence is characterized by progesterone resistance, i.e. re-activation of genes that are suppressed by progesterone).

As discussed in greater detail above (response to comment 2), the “dS1” and “dS2” subsets are stromal cells located in the decidua, but they are not decidualized stromal cells and have a gene expression profile of ESFs rather than DSCs (shown in Vento-Tormo *et al.* Figure 3a). Endometrial mesenchymal stem cells (Munoz-Fernandez et al., 2019) and ESFs (Suryawanshi et al., 2018) are also present in the endometrium during the first trimester and persist until term (Richards et al., 1995; Sakabe et al., 2020). Thus, undifferentiated cells (ESFs) are representative of a population of endometrial stromal cells present at the maternal-fetal interface throughout gestation. This is a very important point, and we thank the reviewers for indicating we did not explain this sufficiently. In the revised manuscript we have made this rationale explicit in the Materials and methods section describing the scRNA-Seq data, and have included the statement “However, whether *HAND2* has functions in other endometrial stromal lineage cells such as ESFs, which persist in the pregnant endometrium till term (Sakabe et al., 2020; Richards et al., 1995; Munoz-Fernandez et al., 2019; Suryawanshi et al., 2018), but have received less attention than DSCs during pregnancy, is unknown” in the Results section describing the knockdown experiment to ensure that the reason we focus on ESFs rather than DSCs is clear.

More importantly, as HAND2 is critical for the identity of these cells, perhaps knockdown triggers a stress response? For example, from the data presented in Figure 3—source data 1 (it would be helpful to add gene names), on of the strongest up-regulated gene upon HAND2 knockdown is BLCAP2 [Log2(FC): 10.2], which encodes a protein that reduces cell growth by stimulating apoptosis.

This is such an interesting question! The answer is beyond the scope of this manuscript, but as recent evidence indicates that decidualization evolved from a stress response (Erkenbrack et al., 2018 [PMID: 30142145]), it is likely that many genes playing a role in ESF/DSC biology, decidualization in particular, are also involved in mediating stress responses. In addition, decidualization induces cell cycle arrest (Logan et al., 2012 [PMID: 22534328]) and BLCAP has been reported to arrest the cell cycle at the G1/S checkpoint (Zhao et al., 2016 [PMID: 26986503]), suggesting that an additional function of HAND2 is to regulate cell cycle progression via BLCAP.

We note, however, that while *HAND2* plays an important role in ESF/DSC biology, *Hand2* KO mice have ESFs and DSCs (albeit dysfunctional ones with respect to specific gene expression), making it unlikely that *HAND2* is required for cell-type identity (Li et al., 2011 [PMID: 21330545]). In addition, *BLCAP* is highly expressed in many tissues (see Author response image 1), meaning that its ability to induce cell cycle arrest/apoptosis and its differential expression upon *HAND2* knockdown do not necessarily indicate that the *HAND2* knockdown triggers either a stress response or apoptosis. Given these data, a mechanistic connection between *HAND2* recruitment into endometrial cell expression and cell cycle regulation is very exciting – we thank the reviewer for pointing out this association and will follow up with additional studies of *BLCAP* in the future.

We have also updated the table to include HUGO gene names in addition to ENSEMBL stable transcript IDs.

**Author response image 1. respfig1:** BLCAP expression in the Human Protein Atlas (HPA) RNA-Seq datasets, with tissues sorted by expression level.

6) The authors illustrated the importance of examining the right cellular state: knockdown HAND2 in T-HESC increases IL15 expression whereas it is well established that HAND2 knockdown in decidual cells decreases IL15 expression. Further, IL15 is strongly induced upon decidualization and previous studies on primary endometrial stromal cells demonstrated that IL15 secretion is undetectable in undifferentiated cells whereas it is abundantly secreted upon decidualization (PMID: 31965050). Thus, we suggest that the should repeat HAND2 KD in decidualizing T-HESC and measure IL15 secretion in both states, with and without HAND2 knockdown, in future experiments.

We agree that cell state is important, but note that several studies have already performed this experiment. Shindoh et al., 2014, showed that KD of *HAND2* in human DSCs abrogated *IL15* expression, whereas most recently Murata et al., 2020, demonstrated that *HAND2* and *IL15* levels significantly increase in the secretory phase of human endometrium, and that HAND2 binds the *IL15* promoter. Thus, the role of *HAND2* in the regulation of IL15 in DSCs is well established. For that reason, we focused on ESFs, again with the logic that ESFs persist in the endometrium till term, but have received less attention than DSCs.

7) Figure 3B – it is unclear what is compared here: genes deregulated upon HAND2 knockdown in T-HESC versus knockdown NR2F2, FOXO1 and GAT2 in decidualized primary cultures? If this is the case, the comparison is not informative as it involves two different cell states. It is surprising that FOSL2 was not included in this analysis.

This is a good point. What we had intended to show was that HAND2 regulates a smaller set of genes than other TFs and factors that mediate decidualization reaction, but agree that comparing KD in ESFs to KD in DSCs is confusing. Therefore, we have removed Figure 3B and the associated text from the Results section.

8) We do not understand the relevance of the experiments described in Figure 5 to the context of gestation length or preterm birth. Trophoblast invasion will have been completed in the second trimester of pregnancy – what is the purpose/message of these experiments? What is the level of IL15 secreted by these cells? Again the T-HESC appear not decidualized – so, what is the relevance to either the midluteal implantation window or gestation?

We did not mean to imply a direct connection between these experiments and either gestation length regulation or preterm birth – those inferences are made through the GWASs of gestation length and birth weight. Instead, we wanted to determine if *HAND2* regulation of *IL15* in ESFs had similar or different effects than *HAND2* regulation of *IL15* in DSCs (for which the *HAND2* -> *IL15* connection is quite clear). As we point out above, ESFs persist in the endometrium during pregnancy until term. Thus, the purpose of these experiments was to use T-HESCs as a model of the persistent ESF population. In our cartoon model figure and discussion we propose that the balance between ESF- and DSC-derived IL15 shifts during pregnancy from the first trimester till term, and that this balance may be important for establishing the window of implantation and regulating gestation length (either directly or through recruitment of immune cells, etc.). We have edited the manuscript to make this logic more clear. In addition, we note that it has been proposed that a “decidual clock” may regulate the successful establishment and maintenance of pregnancy, such that severe decidualization defects lead to infertility, moderate defects lead to recurrent pregnancy loss/recurrent spontaneous abortion, and mild defects lead to preterm birth. We were interpreting our results within the context of this model, but had not made that explicit. We now include a discussion of our results in the light of this model (Norwitz et al., 2015).

*IL15* is expressed in ESFs with TPM=15.73, while in DSCs TPM=67.60, but unfortunately we have not quantified protein expression levels in these cells.

9) What is the evolution of IL15 expression at the maternal-fetal interface? Does it parallel HAND2?

*IL15* is not among the unambiguously reconstructed genes and is dropped from the parsimony analyses.

10) Of the 149 genes that unambiguously evolved endometrial expression why was only HAND2 analyzed? We are not suggesting that each gene be followed up with this level of rigor but would you hypothesize that each of the genes you identified play a role in eutherian reproduction? Or are there other major innovations that some of these genes may be associated with? How frequently would this pattern occur by chance?

The 149 genes that unambiguously evolved endometrial expression were significantly enriched in pathways related to the immune system, as well as human phenotype and biological process GO terms related to regulation of immune system. These data suggest that the recruited genes play immune regulatory roles at the maternal-fetal interface, possibly in establishing maternal immunotolerance (to highlight these observations we have added a new “WordCloud” of enriched terms to Figure 1 – now Figure 1B). It is not clear how to estimate the number of genes one would expect to evolve endometrial expression by chance. We could generate parsimony reconstructions for other tissues to infer if greater or fewer gene expression gain/loss events occurred in the endometrium compared to other tissues. However, because it is unclear what would be an appropriate null model for such a comparison, we have avoided making claims that there is something special about the number of genes that evolved endometrial expression and focused on their functions.

*HAND2* was selected for more detailed analyses because it has been previously shown to be important for silencing estrogen signaling and implantation, which are derived traits in Eutherians. We have added a description of this rationale to the Results section.

11) Figures 2F and 4F – there appears to be a gap in the data points during the third trimester (which looks like it says "thirdr"). Is there still a negative trend if each section is analyzed independently as if they were independent datasets? Aka could this linear trend be composed of two separate trends instead?

Unfortunately, there are no samples in the third trimester (the “thirdr” typo has been corrected). We have explored additional GEO and SRA datasets to determine if there are similar data on gene expression in the decidua throughout gestation that include the third trimester, but as far as we are aware, this is the only such dataset. However, we do not believe that the inference of *HAND2* and *IL15* down-regulation is spurious because of missing third trimester data. For example, we have re-analyzed the data comparing expression of *HAND2* and *IL15* in two groups corresponding to early gestation and term (14-24 vs. 37-40 weeks). The statistical test for significant differences in this case is not based on the linear regression of expression levels by gestation week.

**Author response image 2. respfig2:** 

The Gardner-Altman estimation plots in Author response image 2 show the mean difference between relative *HAND2* expression (left) and *IL15* expression (right) at weeks 14-24 (blue dots) and weeks 37-40 (orange dots) of gestation. The relative expression of each gene for both gestation length groups is plotted on the left axes; the mean difference is plotted on floating axes on the right as a bootstrap sampling distribution. The mean difference is depicted as a black dot; the 95% confidence interval (CI) is indicated as a vertical error bar. The unpaired mean difference between *HAND2* expression at 14-24w and 37-40w is -1.45 [95.0% CI: -1.85 – -0.989]; the *P*-value of the two-sided permutation t-test is 0.0004. The unpaired mean difference between *IL15* expression at 14-24w and 37-40w is -0.816 [95.0% CI: -1.13 – -0.578]; the *P*-value of the two-sided permutation t-test is 0.002. The effect sizes and CIs are reported above as: effect size [CI width: lower bound – upper bound]. 5000 bootstrap samples were taken; the confidence interval is bias-corrected and accelerated. The *P*-values reported are the likelihoods of observing the effect sizes, if the null hypothesis of zero difference is true. For each permutation *P*-value, 5000 reshuffles of the control and test labels were performed.Thus, we believe that our inference that the expression of both *HAND2* and *IL15* decreases throughout gestation is robust, but without data from the third trimester we cannot make a claim about the pattern of decrease. For example, is it gradual from implantation to term, or is there a precipitous decrease during the onset of the third trimester that remains low until term? We have edited the manuscript to reflect this uncertainty – e.g. changing “decreases in expression until term” to “decreases in expression from the first trimester to term”.

12) Please provide the binary encoded data used for this analysis as it could be readily used by other research groups for similar analysis. The custom database of genes implicated in preterm birth would also be a useful dataset.

We have included the binary encoded data along with ancestral state reconstructions (Figure 1— source data 2) and the custom database of genes implicated in preterm birth (Figure 1—source data 9) as source data supplements to Figure 1.

13) It was helpful to hear from one of the authors that the known HAND2 gene wasn't knocked out in mice, so it was an easy early pregnancy gene to start with. Perhaps this should be stated in the revised manuscript?

We apologize if we were unclear. Previous studies have generated mice with a conditional knockout of *Hand2* in uterine tissue (Jones et al., 2013; Li et al., 2011). These mice have implantation defects, thus while the effects of *Hand2* in early pregnancy can be studied, *Hand2* functions in late pregnancy cannot. We have clarified this in the revised manuscript.

14) To reproduce the study, there were a couple of questions around the production of the conditioned media including, how long were the cells incubated in the media and what was the volume of the media use. Please include this information in the revised manuscript.

500μl of media conditioned for 12 hrs – this information has been added to the Materials and methods section.

15) Can you further explain why the opossum was used to measure the estrogen levels?

We used opossum (*Monodelphis domestica*) as a Marsupial model because it lacks maternal recognition of pregnancy and thus is a good representative of the Therian common ancestor. In contrast, other Marsupials such as tammar wallaby have derived traits related to pregnancy, including delayed ovulation and independently evolved maternal recognition. To ensure this rationale is clear, we have expanded on why we selected opossum as a model Marsupial in the Results sections.

16) The relationship between ESR1 and HAND2 is a little unclear. Is ESR1 expression correlated with HAND2 expression in all species studied?

*ESR1* is expressed in the endometria of all species studied. We cannot address whether *ESR1* expression correlated with *HAND2* expression because we don’t have non-pregnant samples from all studied species.